# Selective tubulin-binding drugs induce pericyte phenotype switching and anti-cancer immunity

Bo He [1], Kira H Wood [1], Zhi-jie Li[1,2], Judith A Ermer [1], Ji Li[1], Edward R Bastow [1], Suraj Sakaram [3], Phillip K Darcy[4], Lisa J Spalding [5], Cameron T Redfern[5], Jordi Canes [6], Mafalda Oliveira[6,7,8], Aleix Prat[6,9,10,11], Javier Cortes[6,12,13,14,15,16], Erik W Thompson [17], Bruce A Littlefield[18], Andrew Redfern[5,19] & Ruth Ganss [1]✉

## Abstract

The intratumoral immune milieu is crucial for the success of anti-cancer immunotherapy. We show here that stromal modulation by the tubulin-binding anti-cancer drugs combretastatin A4 (CA-4) and eribulin improved tumor perfusion and anti-tumor immunity. This was achieved by reverting highly proliferative, angiogenic pericytes into a quiescent, contractile state which durably normalized the vascular bed and reduced hypoxia in mouse models of pancreatic neuroendocrine cancer, breast cancer and melanoma. The crucial event in pericyte phenotype switching was RhoA kinase activation, which distinguished CA-4 and eribulin effects from other anti-mitotic drugs such as paclitaxel and vinorelbine. Importantly, eribulin pre-treatment sensitized tumors for adoptive T cell therapy or checkpoint inhibition resulting in effector cell infiltration and better survival outcomes in mice. In breast cancer patients, eribulin neoadjuvant treatment induced pericyte maturity and RhoA kinase activity indicating similar vessel remodeling effects as seen in mice. Moreover, a contractile pericyte signature was associated with overall better survival outcome in two independent breast cancer cohorts. This underscores the potential of re-purposing specific anti-cancer drugs to enable synergistic complementation with emerging immunotherapies.

**Keywords** Angiogenesis; Pericytes; Microtubule-binding Drugs; Tumor Immunity; Rho Kinase

**Subject Categories** Cancer; Immunology; Vascular Biology & Angiogenesis

See also: AL Risinger

## Introduction

Anti-cancer immunotherapies have rapidly expanded in recent years, however, their effectiveness is highly dependent on, and potentially limited by, the tumor immune environment (Galon and Bruni, 2019; Mellman et al, 2023). For instance, the most important predictor for responsiveness to checkpoint blockade is the frequency of intratumoral adaptive immune cells, highlighting the need for optimal T cell numbers in tumors (Cabrita et al, 2020; Helmink et al, 2020; Petitprez et al, 2020; Tumeh et al, 2014). T cell infiltration into solid cancers is controlled by the vasculature; the tumor's abnormal blood vessels co-evolve with an immune-suppressive microenvironment to restrict T cell influx (Dirkx et al, 2003; Griffioen et al, 1996; Hamzah et al, 2008; Motz et al, 2014). Activating tumor blood vessels to express adhesion molecules such as VE-cadherin, ICAM and VCAM can re-establish vascular barrier function and enables successful endothelial-T cell interactions (Johansson-Percival et al, 2018; Johansson-Percival et al, 2015; Tian et al, 2017; Wang et al, 2020; Zhao et al, 2017). Moreover, vessel normalization, a process which creates more organized, homogeneous and better perfused blood vessels, reduces tumor hypoxia to support Th1-driven anti-tumor mechanisms (Ganss et al, 2002; Tian et al, 2017). Vessel normalization involves changes to the vascular bed and improves alignment of endothelial cells and pericytes which are smooth-muscle-like support cells of the microvasculature. Currently, no treatments induce long-lasting vessel normalization. Anti-vascular endothelial growth factor (VEGF) treatment causes transient vessel

[1]Cancer Microenvironment Laboratory, Harry Perkins Institute of Medical Research, Centre for Medical Research, The University of Western Australia, Perth, WA, Australia. [2]Department of Geriatrics and Shenzhen Clinical Research Centre for Geriatrics, Shenzhen People's Hospital (The Second Clinical Medical College, Jinan University, The First Affiliated Hospital, Southern University of Science and Technology), Guangdong, P. R. China. [3]INSiGENe Pty Ltd, UGenome, Tucson, AZ, USA. [4]Cancer Immunology Program, Peter MacCallum Cancer Centre, Melbourne, Victoria, Australia. Sir Peter MacCallum Department of Oncology, The University of Melbourne, Parkville, VIC, Australia. [5]Harry Perkins Institute of Medical Research, Centre for Medical Research, The University of Western Australia, Perth, WA, Australia. [6]SOLTI Cancer Research Group, Barcelona, Spain. [7]Medical Oncology Department, Vall d'Hebron University Hospital, Barcelona, Spain. [8]Breast Cancer Group, Vall d'Hebron Institute of Oncology (VHIO), Barcelona, Spain. [9]Medical Oncology Department, Hospital Clinic of Barcelona, Barcelona, Spain. [10]Translational Genomics and Targeted Therapies in Solid Tumors, August Pi i Sunyer Biomedical Research Institute (IDIBAPS), Barcelona, Spain. [11]Faculty of Medicine and Health Sciences, University of Barcelona, Barcelona, Spain. [12]Medica Scientia Innovation Research (MEDSIR)—Oncoclínicas&Co, Jersey City, NJ, USA. [13]Medica Scientia Innovation Research (MEDSIR)—Oncoclínicas&Co, Sao Paulo, Brazil. [14]International Breast Cancer Center (IBCC), Pangaea Oncology, Quiron Group, Barcelona, Spain. [15]Faculty of Biomedical and Health Sciences, Department of Medicine, Universidad Europea de Madrid, Madrid, Spain. [16]IOB Madrid, Institute of Oncology, Hospital Beata María Ana, Madrid, Spain. [17]School of Biomedical Sciences and Centre for Genomics and Personalised Health, Faculty of Health, Queensland University of Technology (QUT) and Translational Research Institute, Brisbane, Australia. [18]Eisai Inc., Cambridge, MA, USA. [19]Fiona Stanley Hospital, Perth, WA, Australia. ✉E-mail: ganss@perkins.uwa.edu.au

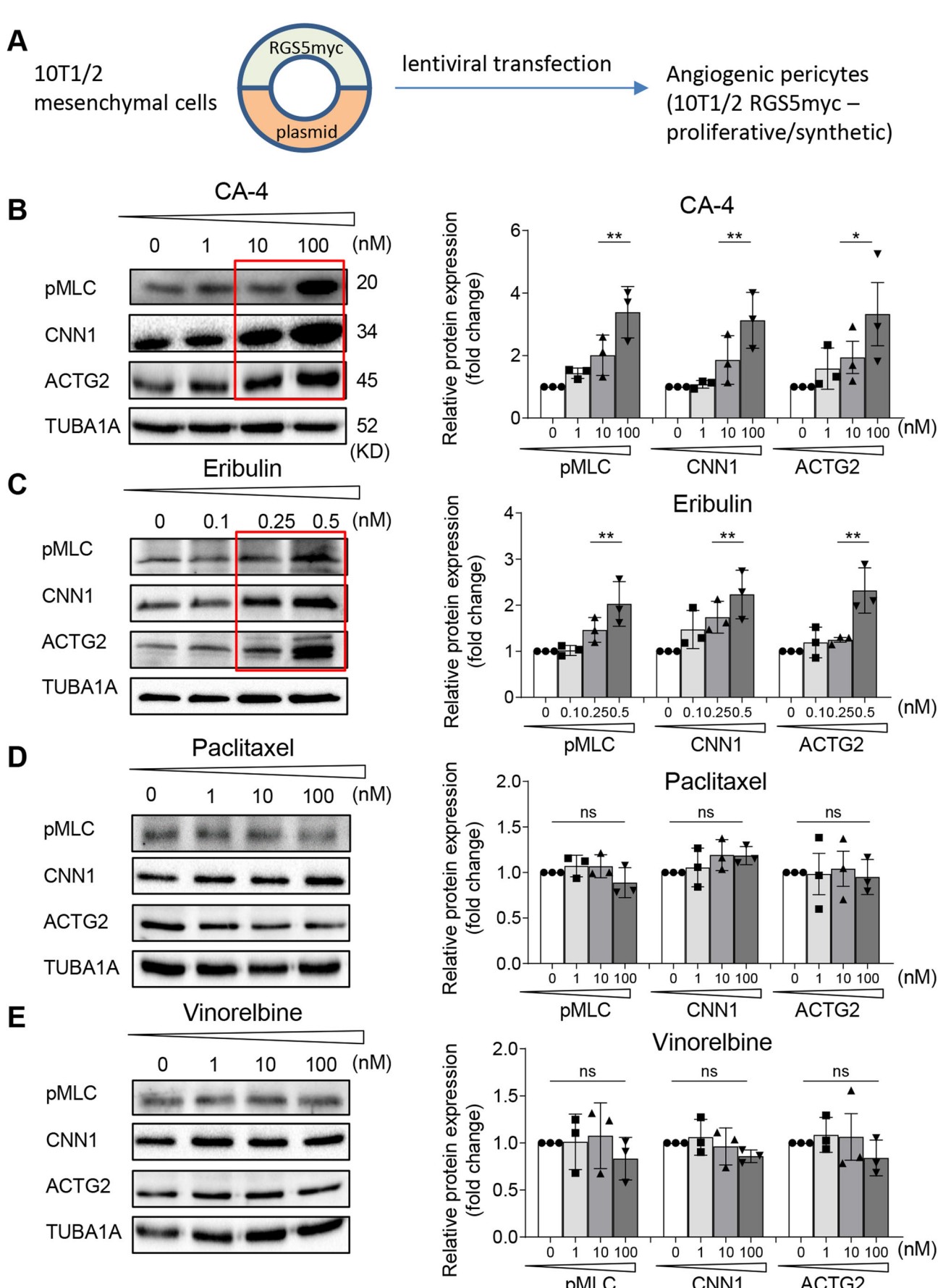

**Figure 1.  Microtubule-binding drugs selectively induce contractile markers in a model of angiogenic pericytes in vitro.**

(A) Generation of a model of "angiogenic" pericytes by overexpressing the Rgs5 gene in mesenchymal 10T1/2 cells using lentiviral technology. (B–E) Representative western blots from 10T1/2 RGS5myc cells incubated with increasing doses of microtubule-binding drugs for 24 h. Contractile pericyte marker (CNN1, ACTG2) intensities in correlation to RhoA kinase activity (pMLC) were quantified by densitometric analysis and normalized on tubulin intensity ($n = 3$ independent experiments). Red boxes highlight upregulation of protein expression. Data were analyzed using one-way ANOVA and expressed as mean ± SEM. (B) CA-4 (**$P = 0.0015$ for pMLC; **$P = 0.0069$ for CNN1; *$P = 0.0329$ for ACTG2, 10 nM CA-4 compared to 100 nM); (C) eribulin (**$P = 0.0054$ for pMLC; **$P = 0.0099$ for CNN1; **$P = 0.0017$ for ACTG2, 0.25 nM eribulin compared to 0.5 nM); (D) paclitaxel and (E) vinorelbine, ns, not statistically significant. Source data are available online for this figure.

normalization (Jain, 2014; Martin et al, 2019), but VEGF blockade ultimately kills blood vessels which then induces hypoxia and adaptive resistance (Casanovas et al, 2005; Paez-Ribes et al, 2009). Tumor pericytes, not just endothelial cells, are requisite targets to effect vessel normalization. Recently, we found that angiogenic tumor pericytes exist in a synthetic, highly proliferative state but can be switched to a more mature (or contractile) phenotype by activating intrinsic Rho-associated coiled-coil forming kinase (ROCK1, RhoA kinase (Johansson-Percival et al, 2015; Li et al, 2024). RhoA kinase acts on the cytoskeleton to regulate the shape, proliferation and movement of pericytes (Kutcher et al, 2007; Li et al, 2024). Potentially, combining drugs which promote pericyte maturation and thus vessel normalization may act in synergy with immunotherapy.

Interestingly, (CA-4), a tubulin binding agent best known as a vascular disrupting drug and currently in clinical development for ovarian and anaplastic thyroid cancers (Grisham et al, 2018), is one of only few compounds known to induce RhoA kinase activity in the cancer vasculature (Williams et al, 2014). However, vascular disrupting agents have disappointing single agent activity due to neuro- and cardiovascular toxicity (Bates and Eastman, 2017) which opens the field to alternative applications. Eribulin mesylate (eribulin, Eisai Inc.), another microtubule inhibitor approved for advanced breast cancer patients previously treated with an anthracycline and a taxane (O'Shaughnessy et al, 2019), has also been associated with vascular remodeling that is qualitatively different to that resulting from anti-angiogenesis therapy (Cortes et al, 2018; Funahashi et al, 2014). Eribulin has shown synergistic effects with immunotherapy in mouse and human cancers suggesting potential vessel normalization mechanisms (Niwa et al, 2023; Tolaney et al, 2021). How eribulin remodels cancer vasculature is currently unknown. Besides eribulin, the tubulin-binding drugs paclitaxel (Taxol) and vinorelbine (Navelbine) are also approved for treatment of metastatic breast cancer (Abu Samaan et al, 2019; Langkjer et al, 2019). When administered in frequent low doses as "metronomic chemotherapy" (Kerbel and Kamen, 2004) they reduce endothelial cell proliferation in vitro and blood vessel density in vivo. These anti-angiogenic stromal effects are multi-facetted and may be unrelated to their anti-mitotic activity (Scharovsky et al, 2009; Wang et al, 2003).

Here we demonstrate that eribulin and CA-4, but not paclitaxel or vinorelbine, induce pericyte phenotype switching in the angiogenic vasculature. Pericyte maturity and contractile marker expression are regulated through opposing roles of RhoA kinase and ERK signaling.

Importantly, low dose treatment regimens improve overall tumor perfusion without killing cancer cells, but act synergistically with immunotherapy. The application of established drugs to achieve novel therapeutic effects such as stromal remodeling creates highly translatable opportunities for therapeutic targeting, in particular for cancers which have been intrinsically resistant to current immunotherapy regimens.

# Results

## CA-4 and eribulin induce differentiation of angiogenic pericytes in vitro

During tumor angiogenesis, new vascular networks arise through extensive cell migration and proliferation. These networks are often fragile and abnormal (Rohlenova et al, 2018). Reducing endothelial or pericyte proliferation re-establishes functionally competent blood vessels (Johansson-Percival et al, 2018; Johansson-Percival et al, 2015). To test the effects of tubulin-binding drugs on highly proliferating pericytes, we used an in vitro model which over-expresses the Regulator of G protein signaling 5 (RGS5) that is highly upregulated in angiogenic tumor pericytes (Hamzah et al, 2008). Mesenchymal 10T1/2 cells were stably transfected with the Rgs5 gene to generate "angiogenic" 10T1/2 RGS5myc cells (Li et al, 2024) (Fig. 1A). Increasing doses of CA-4 or eribulin enhanced RhoA kinase activity (phospho myosin light chain, pMLC, as surrogate marker) and expression of contractile vascular markers such as calponin (CNN1) and (γ)-2 actin (ACTG2) in 10T1/2 RGS5myc cells (Fig. 1B,C). This is indicative of a pericyte phenotype switch from proliferation to quiescence/maturation effected through RhoA kinase activation (Johansson-Percival et al, 2015; Li et al, 2024). Of note, eribulin induced contractile pericyte markers at 40-200 fold lower doses than CA-4. In contrast, neither paclitaxel nor vinorelbine at doses comparable to CA-4 or eribulin increased pMLC or contractile markers (Figs. 1D,E and EV1A). Moreover, RhoA kinase activity was not upregulated in mouse endothelial cells following eribulin or CA-4 treatments at doses which induce pericyte phenotype switching in vitro (Fig. EV1B). These findings suggest highly specific effects of CA-4 and eribulin on pericyte maturation which warranted exploration in angiogenic tumors in vivo.

## CA-4 and eribulin induce pericyte phenotype switching in vivo

Tumor stromal cells such as macrophages and fibroblasts are highly plastic and can differentiate into diverse phenotypes which modulate their functional activities (Goswami et al, 2023; Sahai et al, 2020). Less is known about pericytes, in particular their capacity to initiate and maintain tumor vessel normalization (Hamzah et al, 2008). To assess the potency of microtubule-binding drugs to change pericyte phenotype in vivo, a highly angiogenic mouse pancreatic

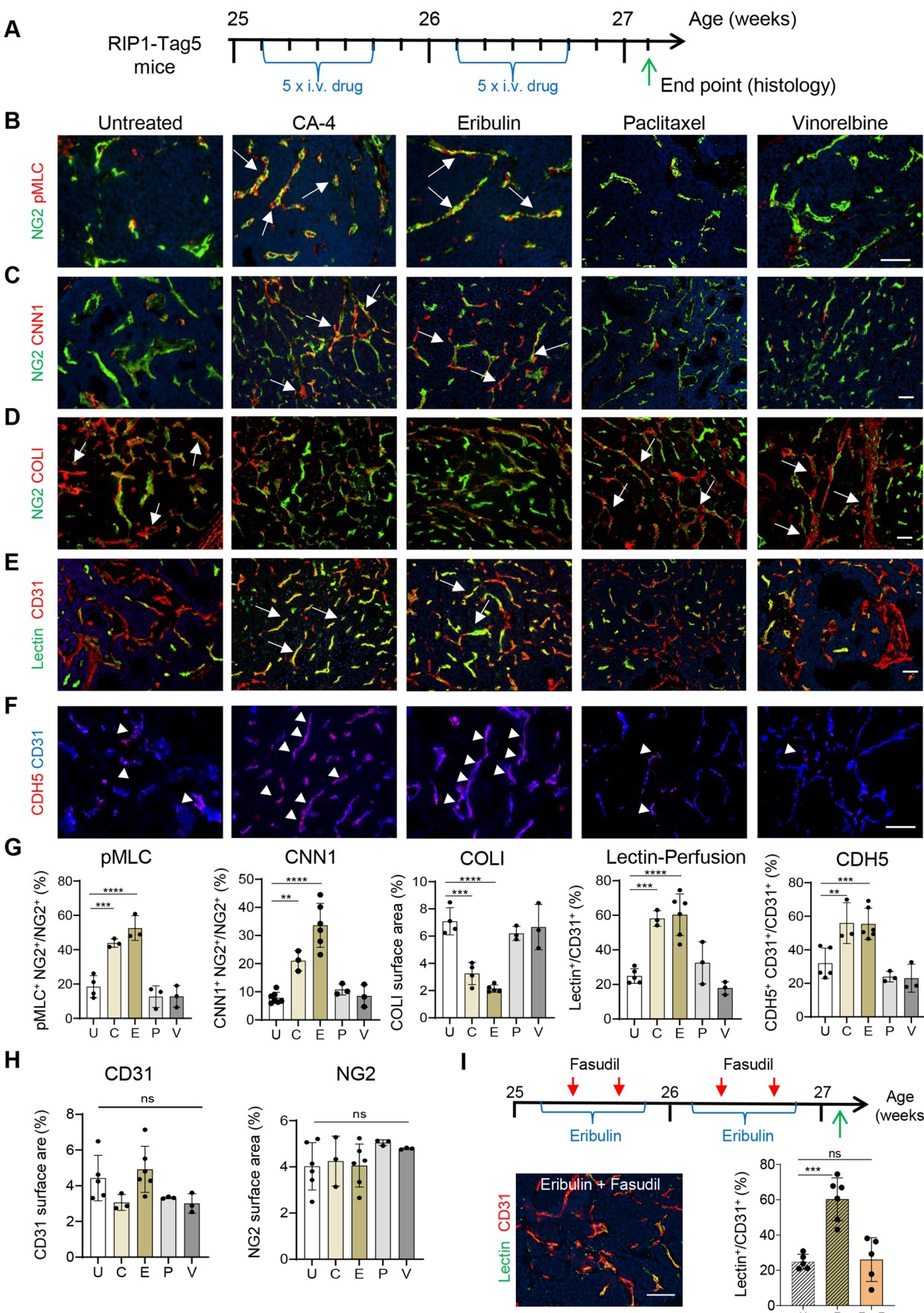

**Figure 2. Microtubule-binding drugs induce pericyte maturity in angiogenic tumor blood vessels in vivo.**

(A) Drug treatment scheme for PNET-bearing RIP1-Tag5 mice (Untreated (U), CA-4 (C), eribulin (E), paclitaxel (P), vinorelbine (V)). (B–F) Representative fluorescence micrographs showing for RIP1-Tag5 tumors: (B) pMLC (red, RhoA kinase activity marker) expression in tumor pericytes (NG2, green); arrows indicate overlay (yellow). Scale bar, 50 μm. (C) Calponin (CNN1, red) expression in tumor pericytes (NG2, green); arrows indicate overlay (yellow). Scale bar, 50 μm. (D) Collagen I (COLI, red) deposition around intratumoral pericytes (NG2, green); arrows indicate COLI deposits. Scale bar, 50 μm. (E) Vascular CD31 expression (red) and infused FITC-lectin (green) as surrogate marker for tumor perfusion, arrows indicate overlay (yellow). Scale bar, 50 μm. (B–E) Counterstained with DAPI (blue). (F) VE-Cadherin (CDH5, red, arrow head) coverage of CD31+ vessels (blue). Scale bar, 50 μm. (G) Quantification of pMLC: $n = 4$ mice in untreated, $n = 3$ mice in all treatment groups, ***$P = 0.0006$, ****$P < 0.0001$; CNN1: $n = 7$ mice in untreated, $n = 6$ mice in eribulin, $n = 3$ mice in all other groups, **$P = 0.0037$, ****$P < 0.0001$; COLI: $n = 4$ mice in untreated and CA-4, $n = 5$ mice in eribulin, $n = 3$ mice in paclitaxel/vinorelbine treatment groups, ***$P = 0.0001$, ****$P < 0.0001$; Lectin-perfusion: $n = 5$ mice in untreated, $n = 6$ in eribulin, $n = 3$ in all other treatment groups, ***$P = 0.0004$, ****$P < 0.0001$; CDH5: $n = 5$ mice in untreated, $n = 6$ mice in eribulin, $n = 3$ mice in all other treatment groups, **$P = 0.0086$, ***$P = 0.0023$. (H) Quantification of CD31+ blood vessels ($n = 5$ mice in untreated, $n = 6$ mice in eribulin, $n = 3$ mice in all other groups) and NG2+ pericytes ($n = 6$ mice in untreated and eribulin, $n = 3$ mice in all other groups), ns, not statistically significant. (I) Eribulin/fasudil double treatment scheme of PNET mice and assessment of tumor perfusion at endpoint (green arrow). A representative fluorescence micrograph of CD31+ vessels (red) and infused FITC-lectin (green) in Eribulin + Fasudil (E + F) treated RIP1-Tag5 tumors is shown. Perfusion (yellow) was quantified in E + F mice in comparison to untreated and eribulin only treated mice (data from G, shadowed), $n = 5$ mice in E + F group, ***$P = 0.0002$, ns, not statistically significant ($P = 0.9748$, U compared to E + F). Scale bar, 50 μm. All data were analyzed using one-way ANOVA. Data are expressed as mean ± SD. Source data are available online for this figure.

neuroendocrine tumor model (PNET, RIP1-Tag5) was employed which is predictive for the efficacy of anti-angiogenic drugs (Ganss and Hanahan, 1998; Nowak-Sliwinska et al, 2018). Tumor-bearing mice were treated with i.v. injections of 1.5 mg/kg CA-4, 0.25 mg/kg eribulin, paclitaxel or vinorelbine for 2 weeks as indicated (Fig. 2A). Importantly, drug dosing was empirically determined for its ability to remodel tumor vasculature without killing blood vessels or cancer cells. Published metronomic chemotherapy at 2 mg/kg for paclitaxel (Chen et al, 2020), induced vessel death and was therefore considered too high for this study (Fig. EV2A). PNET tumors were analyzed by immunohistochemistry for pericyte-specific RhoA kinase activation (pMLC) and expression of contractile (CNN1) and synthetic (collagen I, COLI) pericyte markers. Pericyte-specific expression of pMLC and CNN1 in untreated tumor pericytes was low (Fig. 2B,C). In contrast, substantial COLI deposits around pericytes were evident, indicative of excessive extracellular matrix (ECM) secretion of synthetic pericytes in an angiogenic environment (Fig. 2D). CA-4 or eribulin treatments both induced pericyte pMLC and CNN1 expressions and decreased COLI deposits resembling a contractile, mature pericyte phenotype. Paclitaxel and vinorelbine had no effect on pericyte phenotype in vivo at the applied doses (Fig. 2B–D), which mirrored in vitro results. Following CA-4 or eribulin treatment, PNET tumors were better perfused compared to untreated, paclitaxel- or vinorelbine-treated mice consistent with vessel normalization effects (Fig. 2E,G). Of note, patchy hypervascularity as measured by CD31+ microvessels (vessels/mm²) was evident in some eribulin-treated PNET tumors, but did not result in an overall increase in tumor vascularity (CD31+ or NG2+ surface area), similar to published observations in breast cancer and sarcoma models (Funahashi et al, 2014; Taguchi et al, 2021) (Figs. EV2B and 2H). Moreover, CA-4 or eribulin treatment had significant effects on the entire vascular bed by inducing higher and more continuous expression of the vascular adhesion marker VE-cadherin implying improved barrier function (Fig. 2F,G). To demonstrate a causal role of RhoA kinase in pericyte phenotype switching, PNET-bearing mice were simultaneously treated with eribulin and the RhoA kinase inhibitor fasudil; fasudil suppressed eribulin effects and reduced tumor perfusion to wild type levels (Fig. 2I).

Mechanistically, CA-4 and eribulin both suppressed pERK signaling specifically in tumor pericytes but not endothelial cells (Fig. 3A). In addition, eribulin suppressed pS6R (phospho-S6 ribosomal protein), a downstream target of mTOR signaling and

pAKT in tumor pericytes but not endothelial cells (Figs. 3B and EV3A). Simultaneous suppression of two major signaling pathways may explain why eribulin exerts vascular remodeling effects at a much lower dose than CA-4. Signaling via MEK/ERK and AKT/PI3K are linked to RhoA kinase activity and vascular smooth muscle cell (vSMC) contractility in the periphery, but largely unexplored in the cancer vasculature (Alexander and Owens, 2012; Lacolley et al, 2012). Specific inhibition of ERK and/or PI3K pathways using PD98059 or Wortmannin, respectively, increases contractile marker expression in angiogenic pericytes in vitro (Fig. EV3B) similar to treatment with eribulin or CA-4 (Fig. 1B,C). Simultaneous treatment with both inhibitors increases expression, implying additive effects of these pathways (Fig. EV3B). Moreover, in PNET-bearing mice in vivo blocking of ERK (PD98059) or PI3K (Wortmannin) signaling improves vascular function as measured by increased tumor perfusion (Fig. EV3C). In summary, these data suggest that at low dose, both CA-4 and eribulin normalize tumor blood vessels by inducing pericyte differentiation which in turn improves vessel integrity and overall tumor perfusion. These effects are mediated by opposing roles of RhoA kinase and ERK/AKT signaling.

## CA-4 and eribulin-induced vascular remodeling is highly durable

Stromal remodeling including strategies to normalize tumor vessels are often transient, thereby affording limited opportunity for combination treatments (Jain, 2014). Treatments in the transgenic PNET model, due to its well characterized and slow tumor progression, can be monitored over a sustained period of time to address long-term drug effects beyond the limitations of implantation models (Casanovas et al, 2005). To this end, PNET tumor-bearing mice were treated for 8 weeks with CA-4 or eribulin (Fig. 4A). Treatment with low dose anti-VEGFR (DC101, anti-VEGFR2 monoclonal antibody), known to normalize tumor vessels by targeting endothelial cells, was used in comparison (Huang et al, 2012). Compared to untreated tumors, all drugs significantly improved tumor perfusion which is a functional readout for vessel normalization. However, after prolonged treatment, loss of blood vessels was evident in DC101 treatment groups (Fig. 4B,D). Furthermore, only CA-4 or eribulin, but not DC101 treatment, induced expression of the contractile marker CNN1 in pericytes

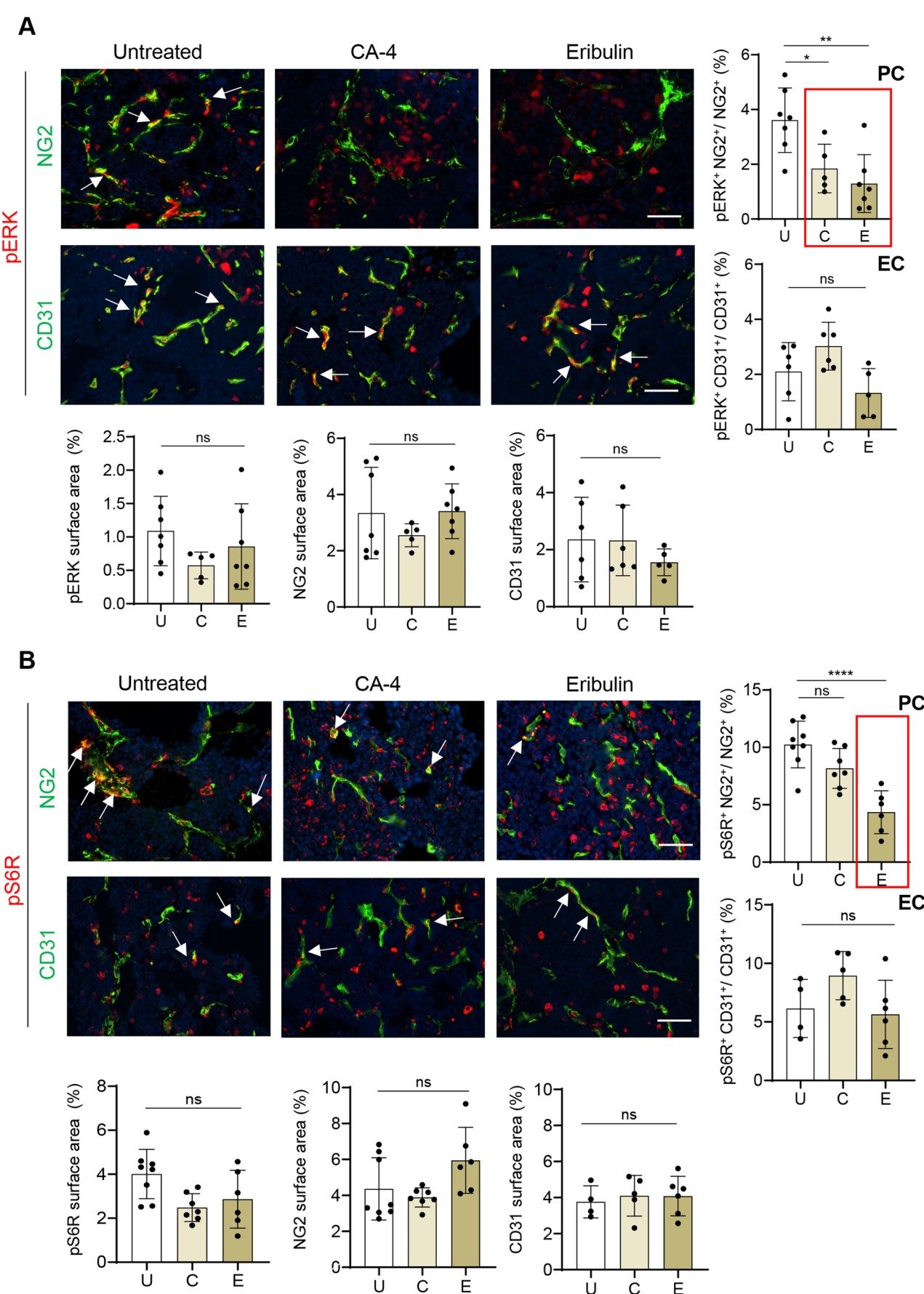

**Figure 3.  CA-4 and eribulin selectively suppress vascular MEK/ERK and pS6R singaling in PNET tumor pericytes in vivo.**

(A) Representative fluorescent micrographs show pERK (red) signals in NG2[+] pericytes (PC, green, upper panels) or CD31[+] endothelial cells (EC, green, lower panels) in untreated (U), CA-4 (C), or eribulin (E) treated RIP1-Tag5 PNET tumors. Arrows indicate overlay (yellow). Quantification of NG2[+] PC or CD31[+] EC specific pERK signals (right), total pERK signals in tumors, total NG2[+] pericytes, and total CD31[+] blood vessels, $n = 7$ mice in untreated (for CD31 analysis $n = 6$), $n = 5$ in CA-4 (for CD31 analysis $n = 6$), $n = 7$ in eribulin (for CD31 analysis $n = 5$) treatment groups, one-way ANOVA. *$P = 0.0225$ (U compared to C), **$P = 0.0017$ (U compared to E), ns, not statistically significant, U compared to C or E. Data are expressed as mean ± SD. Red box emphasizes statistically significant suppression of pERK in PCs. Scale bars, 50 μm.
(B) Representative fluorescent micrographs show pS6R (red, downstream target of mTOR pathway) in NG2[+] PC (green, upper panels) or CD31[+] EC (green, lower panels). Arrows indicate overlay (yellow). Quantification of NG2[+] PC or CD31[+] EC specific pS6R signals (right), total pS6R signals in tumors, total NG2[+] pericytes, and total CD31[+] blood vessels, $n = 8$ mice in untreated (for CD31 analysis $n = 4$), $n = 7$ in CA-4 (for CD31 analysis $n = 5$), $n = 6$ in eribulin treatment groups, one-way ANOVA. ****$P < 0.0001$, U compared to E, ns, not statistically significant ($P = 0.1126$, U compared to C). Data are expressed as mean ± SD. Scale bars, 50 μm. All tissue was counterstained with DAPI. Red box emphasizes statistically significant suppression of pS6R in PCs. Source data are available online for this figure.

(Fig. 4C,D). Quantitative analysis of the collagen matrix surrounding PNET tumors demonstrated that CA-4 or eribulin maintained tumor encapsulation whereas DC101 resulted in rupture of the tumor collagen boundary (Fig. 4E), a first step in the process of local tumor invasion driven by hypoxic pressure following the loss of blood vessels under chronic VEGF inhibition (Paez-Ribes et al, 2009). Thus, pericyte maturity induced by CA-4 or eribulin treatment can be maintained over extended periods of time leading to sustained vessel normalization without vessel death and excessive tumor hypoxia.

## CA-4 and eribulin induce pericyte maturity and normalize blood vessels in breast cancer and melanoma

Tubulin-binding drugs such as eribulin, paclitaxel and vinorelbine are FDA-approved for the treatment of advanced breast cancer patients (Cortes et al, 2018; Funahashi et al, 2014; O'Shaughnessy et al, 2019). To assess the stromal effects of these drugs on breast cancer, a C57BL/6 triple negative breast cancer (TNBC) model was employed which expresses the tumor antigen ovalbumin (AT3-OVA) for advanced immunological analyses in conjunction with congenic anti-OVA T cells (Dushyanthen et al, 2017). Orthotopic tumors were treated with low dose drugs (1.5 mg/kg CA-4, 0.25 mg/kg eribulin, paclitaxel or vinorelbine) as shown in Fig. 5A. Drug dosing in AT3-OVA tumors was chosen to not affect tumor growth or reduce vessel counts. At end stage, tumors were assessed for pericyte differentiation and vascular function. Compared to untreated tumors, CA-4 and eribulin induced pericyte-specific upregulation of pMLC (Fig. 5B,E), accompanied by induction of the contractile marker ACTG2 (Fig. 5C,E) indicating pericyte phenotypic changes similar to the PNET model. This was not observed under treatment with paclitaxel or vinorelbine, consistent with previous in vitro and in vivo data. Moreover, pericyte maturation correlated with an overall increase in tumor perfusion compared to untreated, paclitaxel- or vinorelbine-treated mice (Fig. 5D,F). Although clinical applications of eribulin are thus far limited to advanced breast cancer and liposarcoma, it has shown therapeutic efficacy in human melanoma xenograft models (Asano et al, 2018; Ito et al, 2017). We therefore also employed a B16-OVA (Falo et al, 1995) syngeneic mouse model to investigate pericyte phenotype switching and anti-tumor immunity in a different microenvironment (Fig. EV4A). Similar to PNET and breast cancer, both drugs induce vascular pMLC and ACTG2 expression resulting in enhanced tumor perfusion and reduction of hypoxia (Fig. EV4B–E). Thus, tubulin-binding drugs which induce pericyte

RhoA kinase activity consistently remodel tumor vasculature in diverse mouse tumor types indicative of underlying shared biological principles.

## Eribulin-induced vessel remodeling improves tumor oxygenation, T cell diapedesis and response to immunotherapy

Immunotherapy, in particular checkpoint blockade, is effective in about 50% of advanced melanoma patients (Wolchok et al, 2017), but largely ineffective in breast cancer patients (Winer et al, 2021). In the AT3-OVA breast cancer model, increased perfusion following eribulin treatment resulted in improved tumor oxygenation and vascular ICAM expression (Fig. 6A). To determine the relative contribution of these microenvironmental changes to the tumor immune landscape, intratumoral immune cell profiles were analyzed by flow cytometry. Here, we focussed on eribulin because it is FDA-approved and the tumor vasculature is exquisitely sensitive to low dose drug treatment. Intratumoral adaptive immune cells were profoundly altered in response to eribulin treatment. In particular, total CD4[+] T cell numbers increased concomitantly with a major shift in Ki67[+] GrzB[+] effector CD4[+] T cells and reduced T reg cells (Figs. 6B and EV5A), consistent with an increase in activated CD4[+] T cells in spleens of breast cancer-bearing mice following eribulin treatment (Takahashi-Ruiz et al, 2022). Intratumoral B cells (Fig. 6C) and CD8[+] T cells (Fig. 6D) were also significantly increased. Moreover, amongst CD8[+] T cells, more Ki67[+] GrzB[+] effector T cells and a population of CD69[+] CD103[+] tissue-resident memory T cells (T_{RM}) were induced (Figs. 6D and EV5B); T_{RM} have been associated with improved clinical outcome in TNBC patients and CD103[+] tumor-infiltrating lymphocytes are essential for eribulin-mediated anti-tumor effects in mice (Oya et al, 2023; Virassamy et al, 2023). Following adoptive T cell transfer of pre-activated, congenic H-2K[b]/OVA-specific TCR transgenic T cells (OT-I) (Clarke et al, 2000), significantly more OT-I T cells were detected in eribulin treated than in untreated tumors (Figs. 6E and EV5C). Importantly, combination immunotherapy of eribulin with OT-I T cell adoptive transfer significantly delayed tumor growth, and extended survival when compared to single treatment modalities (Fig. 6F; Appendix Fig. S1). Similarly, eribulin treatment enhanced efficacy of anti-PD-1 checkpoint inhibition and overall survival (Fig. 6G, Appendix Fig. S1). Following eribulin treatment, B16-OVA tumors became highly accessible for adoptively transferred OT-I T cells compared to untreated melanoma (Fig. 7A), consistent with eribulin-induced

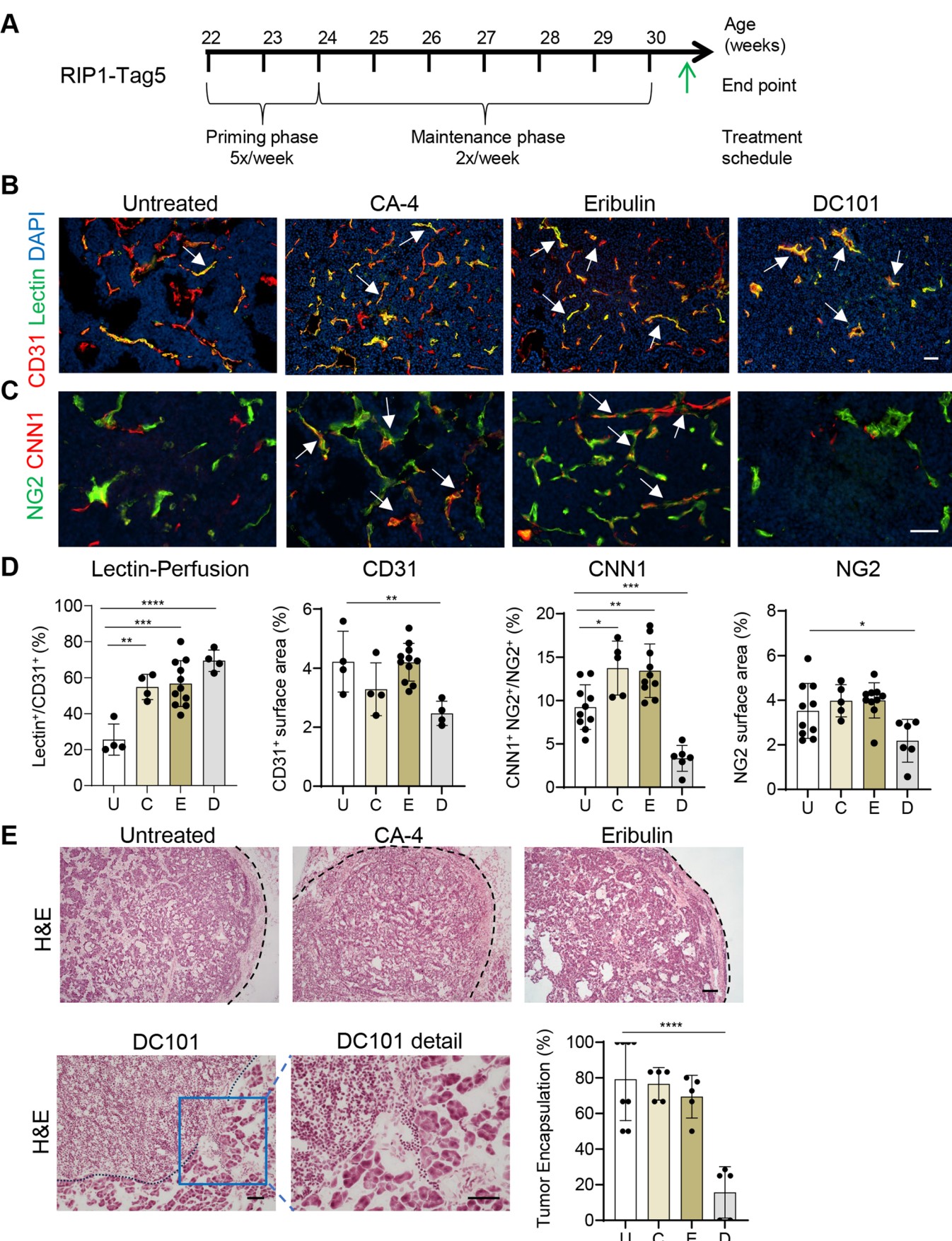

**Figure 4. CA-4 and eribulin induce highly durable vessel normalization effects in PNET.**

(A) 8-week drug treatment scheme in RIP1-Tag5 mice including a 2-week priming and 6-week maintenance phase. (B) Representative fluorescent micrographs show vessel perfusion in untreated mice (U), or mice treated with CA-4 (C), eribulin (E), or anti-VEGFR2 antibodies (DC101, D). FITC-lectin (green) overlay (yellow, marked by arrows) with CD31+ (red) blood vessels is shown. Scale bar, 50 µm. (C) Representative fluorescent micrographs show staining of the contractile marker CNN1 (red) in NG2+ pericytes (green) and overlay (yellow, marked by arrows). Scale bar, 50 µm. (D) Quantification of lectin-perfusion in treatment groups: $n = 4$ mice for untreated, CA-4 and DC101, $n = 11$ mice for erbulin treatment groups, **$P = 0.0024$, ***$P = 0.0002$, ****$P < 0.0001$. CD31+ tumor blood vessels: $n = 4$ mice for untreated, CA-4 and DC101, $n = 11$ mice for erbulin treatment groups, **$P = 0.0081$, DC101 compared to U. CNN1: $n = 10$ mice for untreated and CA-4, $n = 5$ mice for eribulin and $n = 6$ mice for DC101 treatment groups, *$P = 0.0139$, **$P = 0.0048$, ***$P = 0.006$. NG2+ intratumoral pericytes: $n = 10$ mice for untreated and CA-4, $n = 5$ mice for eribulin and $n = 6$ mice for DC101 treatment groups, *$P = 0.0365$, DC101 compared to untreated. (E) Representative H&E micrographs showing tumor cellularity and collagen-rich tumor capsule (black dotted line) in treatment groups. Quantification of percentage of RIP1-Tag5 tumors displaying an intact collagen capsule (dotted line), $n = 8$ mice in untreated, $n = 5$ mice in all treatment groups, ****$P < 0.0001$, DC101 compared to untreated. Scale bar 200 µm (upper images), scale bars for DC101 (lower images), 50 µm. All data were analyzed using one-way ANOVA. Data are expressed as mean ± SD. Source data are available online for this figure.

microenvironmental changes (Fig. EV4). Following adoptive transfers, the M1/M2 ratio of intratumoral macrophages was shifted towards an M1 phenotype in eribulin groups concomitant with an increase in M1 macrophages and a decrease of M2 macrophages reflecting improved tumor oxygenation (Fig. 7B). Consequently, eribulin treatment in combination with adoptive OT-I T cell transfers significantly extended survival (***$P = 0.0007$) compared to untreated, OT-I or eribulin-only treatment groups (Fig. 7C). Overall, at the low dose of 0.25 mg/kg, eribulin did not affect cancer growth per se, but induced stromal changes which alleviated immune suppression and thus sensitized tumors to immunotherapy in mouse breast and melanoma models.

### Eribulin treatment induces pericyte maturity in human breast cancer

To demonstrate that eribulin-induced pericyte phenotype switching can occur in human breast cancer, we employed specimens from the SOLTI-1007 trial of single agent, neoadjuvant eribulin treatment (NCT01669252, 1.4 mg/m² eribulin i.v. on days 1 and 8 every 21-day cycle, for a total of 4 cycles) for HER2-negative breast cancer ($n = 174$; $n = 73$ TNBC, $n = 101$ hormone receptor-positive, HR+) (De Mattos-Arruda et al, 2021; Pascual et al, 2021). This patient cohort enabled analysis of eribulin effects without other pre-treatments. Biopsies at baseline and following the first eribulin cycle (day 15), as well as surgical specimens at 12 weeks (4 eribulin cycles) were analyzed for vascular pericyte coverage, contractile/maturity marker expression in pericytes and overall vessel integrity. Figure 8A demonstrates that PDGFRβ+ pericyte numbers around CD31+ blood vessels remained unchanged during treatment cycles. A change from a "fuzzy" to a sharper vessel appearance was evident over time suggesting closer aligned vascular components following multiple eribulin treatments (Fig. 8A). At baseline (pre-treatment) and after one cycle of eribulin treatment, αSMA and pMLC expression in pericytes was low in both TNBC and HR+ cancers (Fig. 8A). However, following 4 cycles of eribulin treatment peri-vascular expression of the maturity markers αSMA and pMLC was strongly induced in all breast cancers (Fig. 8B,C). Consistently, VE-Cadherin, a marker for endothelial barrier integrity is also upregulated after four eribulin treatment cycles compared to baseline (Fig. 8A–C). These data demonstrate that pericyte phenotype can be modulated by eribulin treatment in a clinical setting similar to findings in mouse cancer models which leads to overall vessel stabilization. Moreover, gene expression data from two breast cancer cohorts (METABRIC $n = 1483$; and BRCA

$n = 1083$) were analyzed for gene signatures associated with pericyte phenotype switching. A contractile/mature pericyte signature positively correlated with longer survival in breast cancer patients in both cohorts (Fig. EV6A,C). In contrast, survival was reduced in correlation with higher expression of synthetic pericyte markers (Fig. EV6B,D). These data suggest that pericyte phenotype is a therapeutic vulnerability in breast cancer and creates opportunities for targeted therapy.

## Discussion

In this study, we demonstrated in mouse tumor models the selective capacity of low dose CA-4 or eribulin to induce tumor pericyte phenotype switching from a highly proliferative to a contractile state by activating RhoA kinase. Vascular remodeling improved tumor oxygenation and alleviated immune suppression which in turn improved immunotherapy efficacy. Moreover, clinical evidence of eribulin-induced pericyte phenotype switching and the prognostic value of pericyte phenotype in breast cancer patient cohorts underscore the therapeutic potential of modulating pericyte plasticity.

In normal organs, pericytes play an important physiological role as guardians of vascular integrity and regulators of blood flow (Dessalles et al, 2021). In cancer, however, pericytes undergo profound morphological changes which foster vessel abnormalities and tumor progression (Hosaka et al, 2016; Meng et al, 2021; Murgai et al, 2017; Wang et al, 2022). Therefore, modulation of aggressive pericyte phenotypes could have profound effects on tumor stroma, and ultimately, on therapeutic outcomes. Indeed, we have previously shown in the transgenic PNET model that highly proliferative pericytes which overexpress RGS5 can be reverted to a more quiescent phenotype by *Rgs5* gene deletion. Changing the property of a single stromal cell population, namely the pericyte, normalized tumor vasculature, facilitated T cell influx and improved anti-cancer vaccination (Hamzah et al, 2008). Subsequently, we identified RhoA kinase (ROCK1) activity as a key determinant of tumor pericyte plasticity (Johansson-Percival et al, 2015; Li et al, 2024).

Here, we aimed to identify drugs which could activate RhoA kinase and thus contractile marker expression in tumor pericytes without causing cancer cell or blood vessel death. The rationale for this approach is underpinned by our previous findings which demonstrated that improved tumor oxygenation—as opposed to hypoxia following blood vessel loss—improves intratumoral

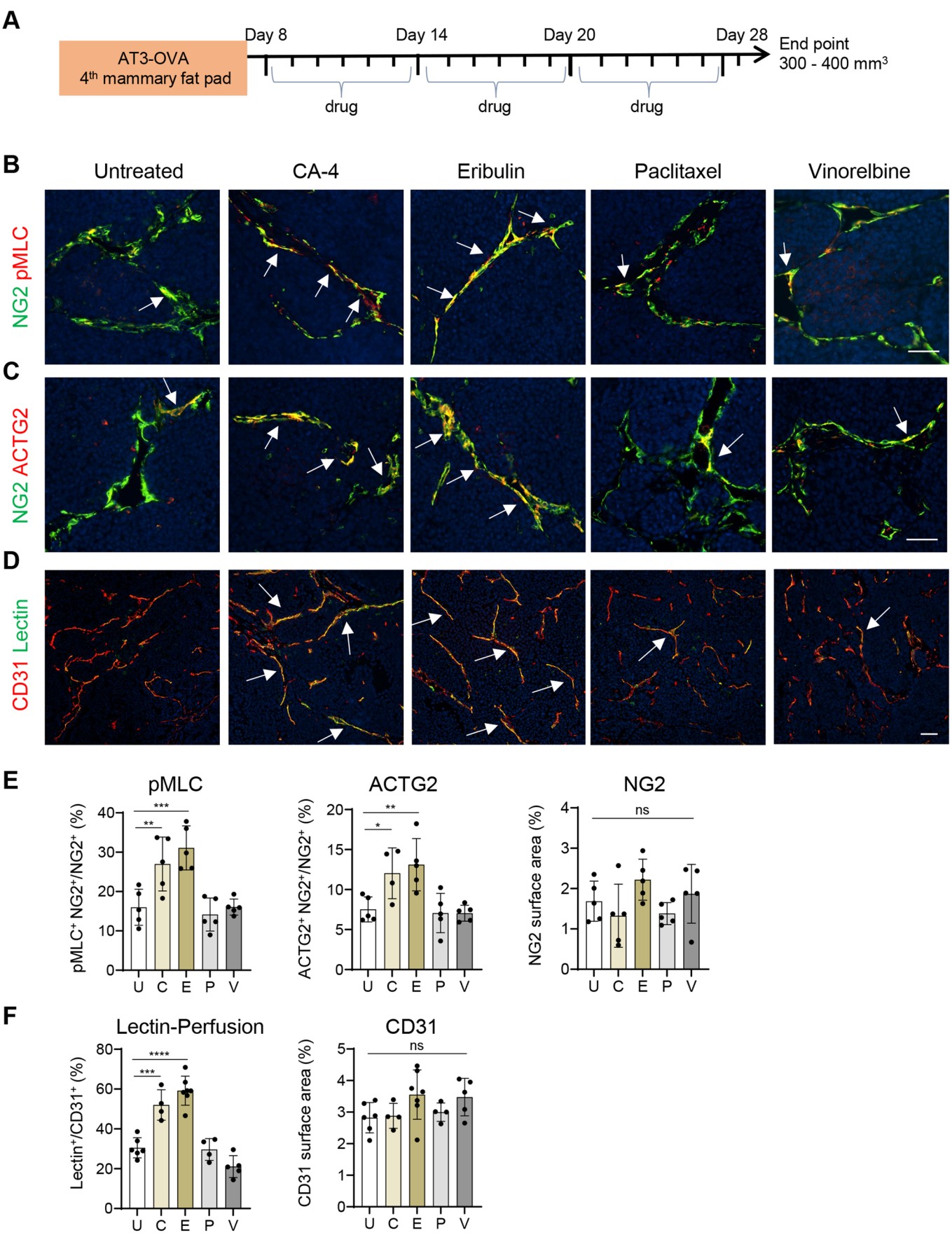

**Figure 5.  CA-4 and eribulin induce pericyte phenotype switching in a triple negative breast cancer model.**

(A) Treatment schedule for the orthotopic TNBC breast cancer model (AT3-OVA) and endpoint for immunohistochemistry. (B) AT3-OVA tumors were grown in untreated C57BL/6 mice (U), or in mice treated with CA-4 (C), eribulin (E), paclitaxel (P) or vinorelbine (V). Representative fluorescent micrographs show pMLC (red) coverage of NG2$^+$ pericytes (green). Arrows point to overlay (yellow). Scale bar, 50 μm. (C) Representative fluorescent micrographs show the contractile marker ACTG2 (red) expression in AT3-OVA intratumoral pericytes (NG2, green). Arrows indicate overlay (yellow). Scale bar, 50 μm. (D) Representative fluorescent micrographs show FITC-lectin (green) overlay (yellow, marked by arrows) with CD31$^+$ (red) blood vessels as surrogate marker for tumor perfusion, Scale bar, 50 μm. (B–D) Counterstained with DAPI (blue). (E) Quantification of pericyte markers (pMLC, ACTG2) and NG2 surface area. pMLC: $n = 5$ mice in all groups, $^{**}P = 0.0072$, $^{***}P = 0.004$; ACTG2: $n = 5$ mice in all groups, $^*P = 0.04$, $^{**}P = 0.006$; NG2: $n = 4$ mice in CA-4, $n = 5$ mice in all other groups, ns, not statistically significant. (F) Quantification of FITC-lectin (green) overlay (yellow) with CD31$^+$ (red) blood vessels and CD31 surface area, $n = 6$ mice in untreated, $n = 7$ mice in eribulin, $n = 4$ mice in CA-4 and paclitaxel, $n = 5$ mice in vinorelbine treatment groups, $^{***}P = 0.0001$, $^{****}P<0.0001$, ns, not statistically significant. All data were analyzed using one-way ANOVA. Data are expressed as mean ± SD. Source data are available online for this figure.

immune function (Hamzah et al, 2008; He et al, 2018; He et al, 2020; Johansson-Percival et al, 2015; Johansson et al, 2012).

Microtubule-binding drugs such as CA-4, eribulin, paclitaxel and vinorelbine have all been evaluated for their effects on tumor vasculature (Funahashi et al, 2014; Grisham et al, 2018; Kerbel and Kamen, 2004). The tubulin stabilizer paclitaxel and destabilizer vinorelbine are in clinical trials in metronomic regimens (Cazzaniga et al, 2022). Metronomic chemotherapy which corresponds to paclitaxel and vinorelbine doses of 2–6 mg/kg in mice (Chen et al, 2020) has anti-angiogenic effects and reduces tumor blood vessel numbers (Bocci et al, 2013; Qin et al, 2018). Consistently, we found that paclitaxel at metronomic dose (2 mg/kg) reduced blood vessels over time. Doses which do not reduce vessel count (0.05, 0.25, 0.5 mg/kg) neither increased tumor perfusion nor triggered pericyte phenotype switching.

Of the four drugs tested in this study, only CA-4 and eribulin induced RhoA kinase activity in tumor pericytes suggesting a shared, non-mitotic pathway which we identified as vascular MEK/ERK suppression. High doses of CA-4 (e.g. 30–100 mg/kg) in mouse tumors cause selective vascular death within hours after treatment (Dark et al, 1997); this also translates into significant reduction of tumor perfusion within 24 h as measured by dynamic contrast enhanced magnetic resonance imaging (DCE-MRI) in rat and human tumors, indicative of anti-vascular effects (Galbraith et al, 2003). Employing low dose CA-4 treatment (1.5–2.5 mg/kg), we demonstrated here that CA-4 induces RhoA kinase activity in angiogenic pericytes in vitro and in vivo leading to durable normalization of the entire vascular bed. We thus provide mechanistic insights which may explain an earlier study where low dose CA-4 application (2 mg/kg) in a rat model of intrahepatic colon carcinoma acted synergistically with anti-cancer immunization without changing the tumor's vascular density (Badn et al, 2006). Low dose CA-4 application therefore warrants further investigations in combination immunotherapies.

Eribulin has well documented vascular remodeling capabilities which appear different to other microtubule binding drugs (Cortes et al, 2018). However, it remains unclear how eribulin modulates tumor vasculature. Here, we demonstrated that eribulin treatment (0.25 mg/kg, approximately twofold lower than the clinical dose of 1.4 mg/m²) (Nair and Jacob, 2016) induced pericyte phenotype switching involving activation of RhoA kinase and downregulation of both MEK/ERK and AKT signaling pathways. Thus, our in vitro and in vivo data now establish pericytes as major targets for eribulin-mediated vascular remodeling. This is consistent with previous in vitro data showing that eribulin but not paclitaxel changed the gene expression profile in primary human brain pericytes (Agoulnik et al, 2014).

How microtubule-targeting drugs induce pericyte phenotype switching involving RhoA kinase and ERK and/or AKT signaling remains largely unknown. As shown in cancer cells, microtubule destabilizers can activate RhoA kinase via the microtubule-associated RhoGEF factor GEF-H1 (Krendel et al, 2002). This in turn may lead to ERK inactivation (Stone and Chambers, 2000). Alternatively, microtubule targeting may directly downregulate ERK and/or AKT signaling leading to RhoA kinase activation (Jo et al, 2014; Nazmy et al, 2023; Wolfrum et al, 2004). In vascular smooth muscle cells, key regulatory kinases such as RhoA kinase, MEK/ERK, and PI3K/AKT modulate phenotype directly via the actin cytoskeleton and at the transcriptional level (Lacolley et al, 2012).

Due to their close proximity and reciprocal regulation, changes in pericytes thus indirectly affect endothelial cells and the entire vascular bed. Indeed, RhoA kinase signaling in vitro controls pericyte shape and contractile properties, and also modulates endothelial cell proliferation (Kutcher et al, 2007). Consistently, as shown here, pericyte-endothelial adhesion was improved with eribulin treatment as assessed by increased and better aligned VE-cadherin signals compared to untreated tumors. Importantly, the vessel remodeling effects are durable and sustained over 8 weeks in clear contrast to chronic low dose anti-VEGFR treatment, thus differentiating eribulin-induced vessel remodeling from VEGF blockade. Interestingly, in advanced breast cancer patients, reoxygenation of tumor tissue was higher on day 7 following eribulin infusion compared to anti-VEGF antibody (bevacizumab) treatment as measured by diffuse optical spectroscopic imaging (DOSI) (Ueda et al, 2016).

We also show that eribulin pre-treatment increased perfusion, and facilitated spontaneous and adoptively transferred T cell infiltration into TNBC which is likely assisted by increased vascular ICAM expression. Tumor vessel activation may be a direct eribulin effect on endothelial cells (Abbona et al, 2022) and/or a result of changes in the inflammatory tumor milieu. Furthermore, pericyte-mediated vessel normalization substantially improved tumor oxygenation, intratumoral immune cell function and T cell infiltration in breast cancer and melanoma models, consistent with clinical observations that eribulin treatment may improve immune status of responders, in particular in TNBC patients (Goto et al, 2022; Kashiwagi et al, 2017).

Using clinical specimens from the SOLTI eribulin neoadjuvant trial, we provided proof-of-concept that activation of αSMA and RhoA kinase in intratumoral pericytes and increased vessel integrity are a direct result of eribulin neoadjuvant treatment in TBNC and HR$^+$ breast cancer patients. Whilst eribulin treatment is

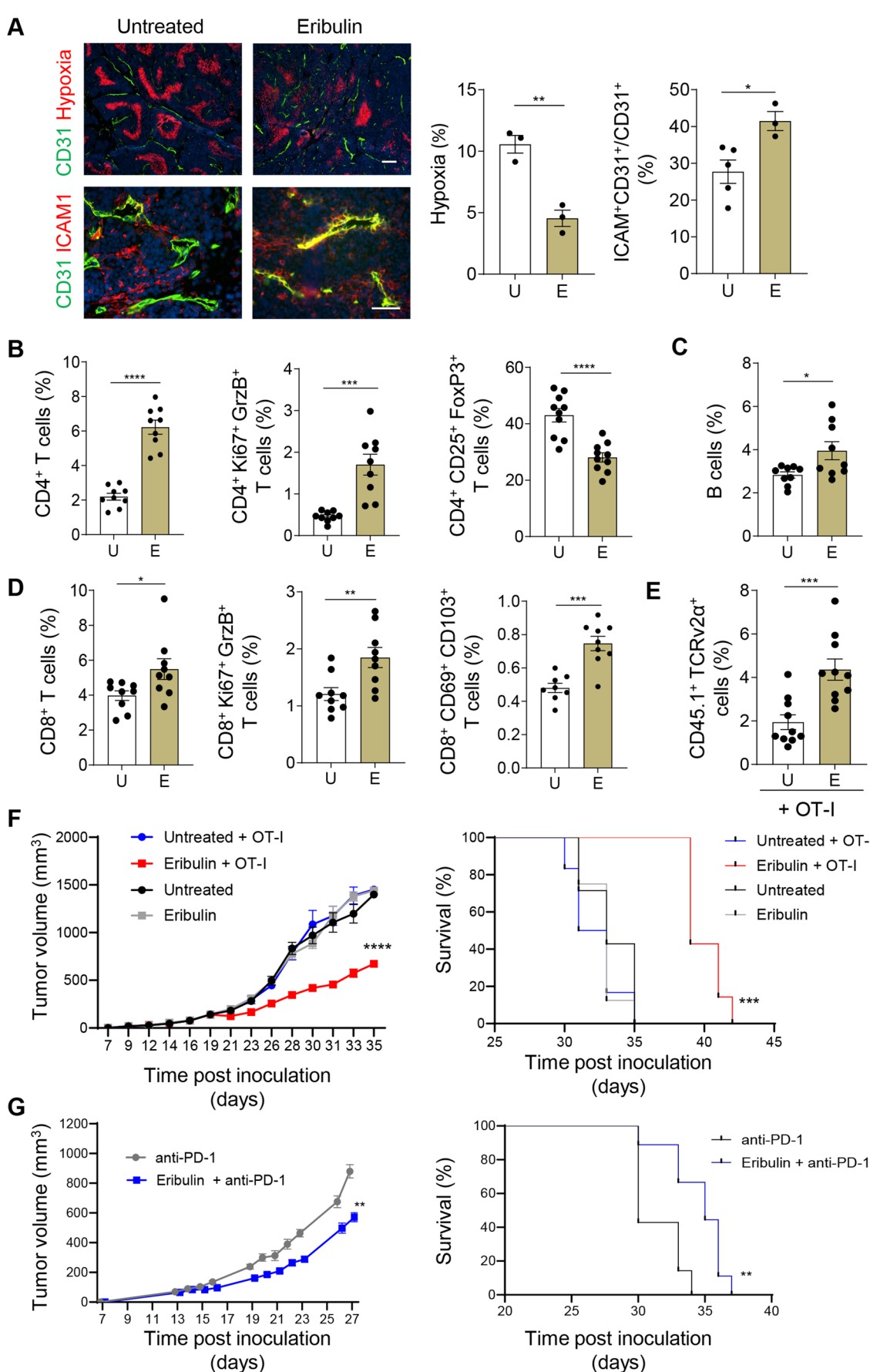

**Figure 6. Eribulin treatment improves effectiveness of anti-cancer immunotherapy in a TNBC model.**

(A) Representative fluorescent micrographs show AT3-OVA tumors grown in C57BL/6 wild type mice untreated (U), or treated with eribulin (E). Tumor hypoxia was visualized with antibodies against pimonidazole adducts and vessel activation with vascular ICAM staining. Hypoxic tumor areas are depicted as red clusters (upper panels) and were quantified (left). Staining of vascular (CD31, green) ICAM (red) expression (lower panels) and quantification of overlay (yellow, right). Hypoxia: $n = 3$ mice in all groups, $**P = 0.0035$; ICAM: $n = 5$ mice for untreated and $n = 3$ mice for eribulin treated groups, $*P = 0.025$. Scale bars, 50 µm. (B) FACS quantification of AT3-OVA intratumoral CD4$^+$ T cells, CD4$^+$ Ki67$^+$ GrzB$^+$ effector T cells and CD4$^+$ CD25$^+$ FoxP3$^+$ T reg in untreated and eribulin treated groups, $n = 9$ mice, $***P = 0.0002$, $****P < 0.0001$. (C) FACS quantification of intratumoral B220$^+$ B cells in untreated and eribulin treated breast cancers, $n = 9$ mice, $*P = 0.0221$. (D) FACS quantification of intratumoral CD8$^+$ T cells, CD8$^+$ Ki67$^+$ GrzB$^+$ effector T cells and CD8$^+$ CD69$^+$ CD103$^+$ T$_{RM}$ cells in untreated and eribulin treated AT3-OVA breast cancers, $n = 8$ mice for T$_{RM}$ analysis in untreated, all other groups $n = 9$ mice, $*P = 0.0337$, $**P = 0.0074$, $***P = 0.0001$. (E) Quantification of intratumoral OT-I CD8$^+$ T cells (CD45.1$^+$ TCRv2α$^+$), gated on live cells, following adoptive OT-I transfer in eribulin treated compared to untreated mice, $n = 10$ mice, $***P = 0.0007$. (A–E) Data were analyzed using a two-tailed, unpaired Student's $t$ test. Data are expressed as mean ± SEM. For FACS gating strategies, see Fig. EV5. (F) AT3-OVA bearing C57BL/6 mice were left untreated ($n = 8$), treated with eribulin ($n = 8$), or treated with adoptive OT-I T cell transfers with ($n = 8$) or without eribulin co-treatment ($n = 6$) every 3rd day from day 17 to 29. Tumor growth curve (left), $****P ≤ 0.0001$, two-way ANOVA, mean ± SEM. Survival (right), $***P = 0.0001$, log rank (Mantel-Cox) test. Individual tumor growth curves are shown in Appendix Fig. S1A. (G) AT3-OVA bearing C57BL/6 mice were treated with anti-PD-1 only ($n = 8$), or with eribulin followed by checkpoint inhibitor treatment ($n = 10$) every 3rd day from day 14-27. Tumor growth (left), $**P = 0.0044$, two-tailed, paired $t$ test, mean ± SEM; Survival (right), $**P = 0.0044$, log rank (Mantel-Cox) test. Individual tumor growth curves are shown in Appendix Fig. S1B. Source data are available online for this figure.

likely to also affect cancer cells, our findings underscore the importance of eribulin-induced vascular changes in promoting tumor oxygenation which potentially contributes to the intrinsic changes to a less aggressive cancer subtype in one third of SOLTI-1007 patients (De Mattos-Arruda et al, 2021; Pascual et al, 2021). This finding is also consistent with a positive correlation of pericyte contractile marker expression and survival in two different breast cancer patient cohorts.

Overall, we demonstrated that low dose eribulin pre-treatment which does not affect tumor growth per se, significantly enhanced immunotherapies by prolonging survival, suggesting that both eribulin dose and application sequence might be important parameters to consider in future immune combination trials.

# Methods

### Reagents and tools table

| Reagent/resource | Reference or source | Identifier or catalog number |
| --- | --- | --- |
| **Experimental models** | | |
| C3H10T1/2 RGS5myc cells (*M. musculus*) | Li et al (2024) | Corresponding author |
| B16-OVA cells (*M. musculus*) | Falo et al (1995) | Prof. G. Hämmerling, DKFZ, Heidelberg, Germany |
| AT3-OVA cells (*M. musculus*) | Dushyanthen et al (2017) | Prof. P. Darcy, Peter MacCallum Centre, Melbourne, Australia |
| SVEC4-10 (*M. musculus*) | ATCC | CRL-2181 |
| B57BL6/J mice (*M. musculus*) | Ozgene Pty Ltd | C57BL/6JOzarc |
| RIP1-Tag5 mice (*M. musculus*) | Ganss and Hanahan (1998) | Prof. Douglas Hanahan, ISREC, Lausanne, Switzerland |
| OT-I x CD45.1 mice (*M. musculus*) | Clarke et al (2000) | Prof. B. Marshall, UWA, Perth, Australia |
| Breast cancer biopsies (*H. sapiens*) | Pascual et al (2021) | NCT01669252 |
| **Recombinant DNA** | | |
| Not applicable | | |

| Reagent/resource | Reference or source | Identifier or catalog number |
| --- | --- | --- |
| **Antibodies** | | |
| Antibodies for Western blots and FACS | This study | Appendix Table S2 |
| Antibodies for Immunohistochemistry and Secondary Antibodies | This study | Appendix Table S3 |
| Anti-VEGFR (*M. musculus*) | BioXCell | DC101 |
| Anti-PD1 (*M. musculus*) | BioXCell | BE0146 |
| Fc-block (CD16/CD32) (*M. musculus*) | BioXCell | 2.4G2 |
| **Oligonucleotides and other sequence-based reagents** | | |
| OVA peptide (SIINFEKL) | Genscript | RP10611 |
| **Chemicals, enzymes and other reagents** | | |
| DMEM, low glucose | ThermoFisher | 11960069 |
| Penicillin-Streptomycin | ThermoFisher | 15140122 |
| L-glutamine | ThermoFisher | A2916801 |
| Combretastatin-A4 | Selleck Chemical | S7783 |
| PEG300 | Sigma | 202371-250 |
| DMSO | Sigma | D5870 |
| Paclitaxel | AdipoGen Life Sciences | AG-CN2-0045 |
| Vinorelbine | AdooQ Bioscience | A10976 |
| Eribulin | Eisai Inc. | n/a |
| Fasudil | LKT Laboratories | F0275 |
| PD98059 | Cayman Chemical | 10006726 |
| Wortmannin | Selleck Chemical | S2758 |
| Lectin-FITC (*Lycopersicon esculentum*) | Vector | FL-1171 |
| Hypoxyprobe Omni kit | Hypoxyprobe Inc. | HP3 |

| Reagent/resource | Reference or source | Identifier or catalog number |
|---|---|---|
| Protease and phosphatase inhibitor cocktail | Sigma | PPC1010 |
| BCA Kit | ThermoFishcer | 23225 |
| Hematoxylin solution | Sigma | GHS332 |
| Eosin Y solution | Sigma | HT110132 |
| Amersham ECL Prime Western Blotting Detection Reagent | Cytiva Life Sciences | RPN2236 |
| OCT compound | ProsciTech | IA012 |
| Recombinant IL2 (*H. sapiens*) | PeproTech | 200-02 |
| Collagenase IV | Worthington | LS004188 |
| DNase I | Worthington | LS002140 |
| ViaDye Violet Fixable Viability Dy Kit | Cytek | SKUR7-60009 |
| True-Nuclear Transcription Factor Buffer Set | BioLegend | 424401 |
| MOM kit | Vector | FMK-2201 |
| AlexaFluor™ 488 Tyramide Superboost Kit, streptavidin | ThermoFisher | B40932 |
| AlexaFluor™ 594 Tyramide Superboost Kit, goat-anti-rabbit IgG | ThermoFisher | B40925 |
| Vectashield mounting medium/DAPI | Vector | H1200 |
| **Software** | | |
| GraphPad Prism 9 and 10 | GraphPad | https://www.graphpad.com/scientific-software/prism/ |
| ChemiDoc™ Imaging System, 6.1 | Bio-Rad | https://www.bio-rad.com/en-au/product/chemidoc-mp-imaging-system?ID=NINJ8ZE8Z |
| SpectroFlo 3.3.0 | Cytek | https://cytekbio.com/blogs/resources/spectroflo |
| Nikon NIS-Elements software Version 4.0 | Nikon | https://www.nikon.com/products/microscope-solutions/support/download/software/imgsfw/nis-f_v4600064.htm |
| Edge R 3.40.2 | Bioconductor | https://bioconductor.org/packages/release/bioc/html/edgeR.html |
| GSVA R 1.46.0 | Bioconductor | https://bioconductor.org/packages/devel/bioc/vignettes/GSVA/inst/doc/GSVA.html |
| Maxstat R package 0.7-25 | The comprehensive R Archive Network | https://cran.r-project.org/web/packages/maxstat/index.html |
| Survminer R package 0.4.9 | The comprehensive R Archive Network | https://cran.r-project.org/web/packages/survminer/index.html |

| Reagent/resource | Reference or source | Identifier or catalog number |
|---|---|---|
| R software 4.2.2 | The comprehensive R Archive Network | https://cran.r-project.org/bin/windows/base/ |
| **Other** | | |
| Polyvinylidene difluoride (PVDF) membrane | Millipore | IPVH00010 |

## Patients and human tissue

Primary breast cancer specimens from female patients were obtained from the phase II study, open-label, single-arm SOLTI-1007 NEOERIBULIN (NCT01669252). SOLTI-1007 evaluated a single-agent (eribulin) neoadjuvant treatment for stage I-II HER2-negative breast cancers, including both hormone receptor positive (HR+) and triple negative breast cancer (TNBC) patients. Patients were treated with a dose of 1.4 mg/m² eribulin intravenously (i.v.) on days 1 and 8 every 21-day cycle for 4 cycles (Pascual et al, 2021). Formalin-fixed baseline biopsies (day 0), biopsies following the first eribulin cycle (day 15) and surgical breast cancer samples after 4 eribulin cycles (week 12) were analyzed from a maximum of 8 HR+ and TBNC randomly selected SOLTI-007 patients (Appendix Table S1). Informed consent was obtained from all patients and the experiments conformed to the principles set out in the WMA Declaration of Helsinki and the Department of Health and Human Services Belmont Report. Analysis of archival human breast cancer tissue was approved by the Royal Perth Hospital Human Research Ethics Committee, Western Australia, Australia (RGS0000001981).

## Mice

RIP1-Tag5 mice express SV40 Large T antigen under the control of the rat insulin promoter and were bred on a C3HeBFe (C3H) background as previously described (Ganss and Hanahan, 1998). C57BL/6 (C57BL/6JOzarc) were purchased from Ozgene Pty Ltd, Perth, Australia. Congenic (CD45.1) H-2Kb-restricted, OVA-specific TCR transgenic (OT-I) mice (Clarke et al, 2000) were a gift from B. Marshall (Perth, Australia). All mice were kept in pathogen-free facilities at the University of Western Australia (UWA) with food and water provided ad libitum. Males and females at 8-12 weeks were randomly assigned for tumor studies and similar findings are reported for both sexes; orthotopic breast cancer studies used female mice only. All animal studies were approved by the UWA Animal Ethics Committee (protocols ET0000455, ET0000492). Studies were not conducted blindly.

## Cell lines

Murine C3H10T1/2 (10T1/2) cells were purchased from the American Type Culture Collection (ATCC), and transfected with RGS5myc using lentiviral technology to generate a model for angiogenic pericytes (10T1/2 RGS5myc) as published (Li et al, 2024). B16-OVA cells (a gift from G. Hämmerling, Heidelberg, Germany) are C57BL/6 murine B16 melanoma cells transfected with ovalbumin (MO4) (Falo et al, 1995). AT3-OVA cells are

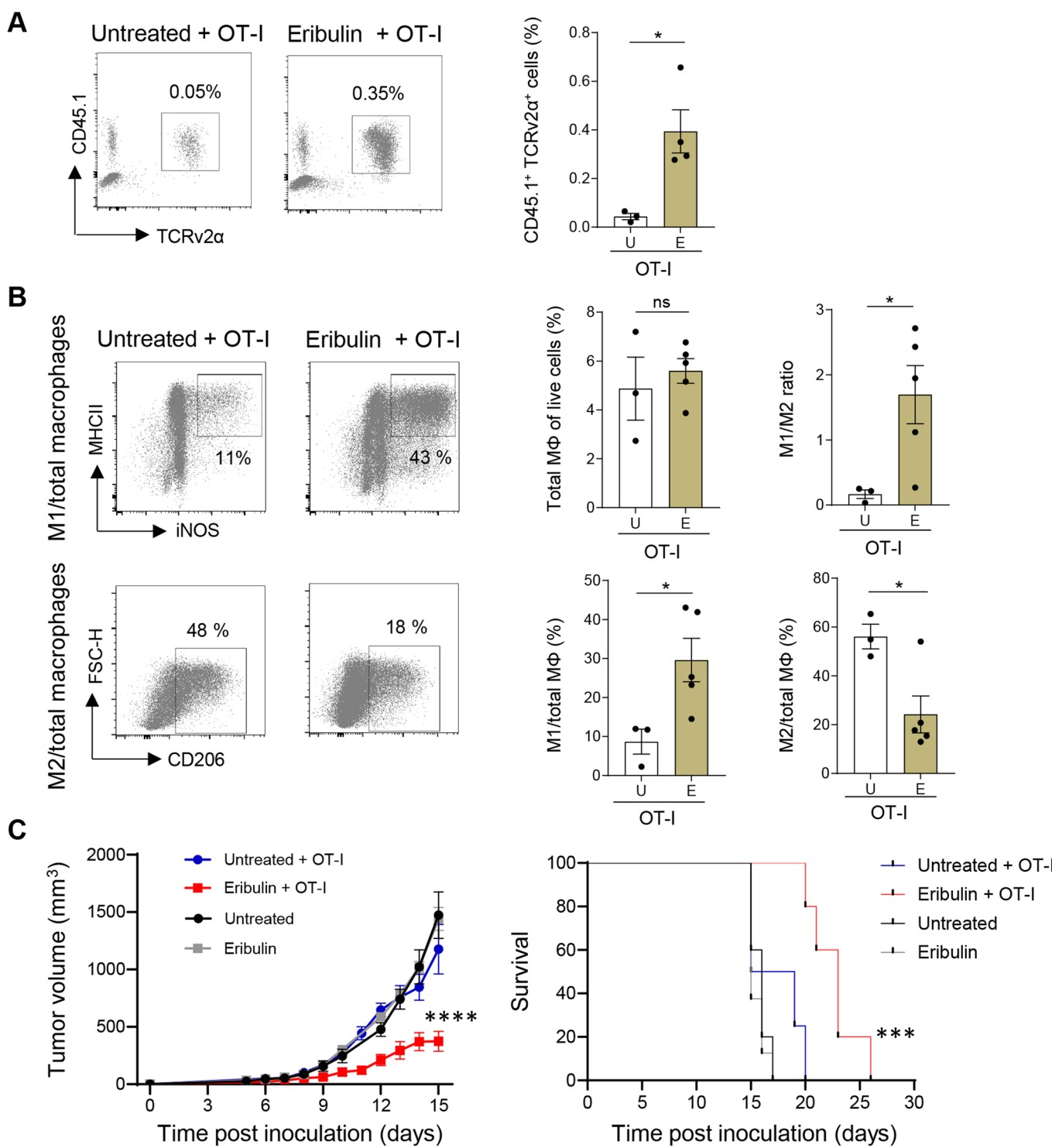

C57BL/6 murine triple negative breast cancer cells transfected with ovalbumin (Dushyanthen et al, 2017). SVEC4-10 (SVEC, ATCC) are SV40-transformed mouse endothelial cells. All cells were cultured in DMEM medium supplemented with 10% FCS, and 100 units/ml penicillin/100 μg/ml streptomycin, and 2 mM L-glutamine. Cell lines were not authenticated as they were newly obtained from vendors/collaborators. All cells were mycoplasma free.

## Drug treatments

Combretastatin (CA-4, S7783, 1.5–2.5 mg/kg in 5% DMSO, 10% PEG300 and 85% PBS, Selleck Chemical), paclitaxel (AG-CN2-0045, 0.05, 0.25, 0.5 or 2 mg/kg in 5% DMSO, 95% PBS, AdipoGen Life Sciences), vinorelbine (A10976, Navelbine, 0.25 mg/kg in 5% DMSO, 95% PBS, AdooQ Bioscience), eribulin (0.05 and 0.25 mg/kg in 5%

◄ **Figure 7. Eribulin treatment increases adoptively transferred effector T cells and M1 macrophage polarization in the melanoma microenvironment.**

(A) Representative FACS blots show gating and quantification of adoptively transferred OT-I T cells (CD45.1$^+$ TCRv2α$^+$) in untreated (U) or eribulin (E, 0.25 mg/kg) treated B16-OVA melanoma, $n = 3$ mice in untreated and $n = 4$ mice in eribulin treated groups, *$P = 0.021$. (B) Representative FACS blots show gating and quantification of intratumoral total macrophages (CD45$^+$, GR1$^+$, CD11b$^+$, F4/80$^+$) in B16-OVA tumors following OT-I adoptive T cell transfers, ns: not statistically significant; shift in M1/M2 ratio, *$P = 0.043$; M1 macrophages (MHCII$^{high}$, iNOS$^+$), *$P = 0.036$; M2 macrophages (CD206$^{high}$), *$P = 0.025$, $n = 3$ mice in untreated and $n = 5$ mice in eribulin treated groups. (A, B) Data were analyzed using two-tailed, unpaired Student's $t$ test and presented as mean ± SEM. (C) B16-OVA bearing mice were left untreated ($n = 5$), treated with 7 doses of eribulin ($n = 8$), or received adoptive OT-I cell transfers with ($n = 5$) or without eribulin co-treatment ($n = 4$) on days 9 and 12. Tumor growth and survival was monitored. Tumor growth (left), ****$P < 0.0001$, two-way ANOVA, mean ± SEM. Survival (right), ***$P = 0.0007$, log rank (Mantel-Cox) test. Individual tumor growth curves are shown in Appendix Fig. S1C. Source data are available online for this figure.

DMSO, 95% PBS, Eisai Inc.) were injected i.v.; for VEGF blocking studies, anti-VEGFR2 antibodies (DC101, 15 mg/kg in PBS, BioXCell) was injected intraperitoneally (i.p.). RhoA kinase inhibitor fasudil hydrochloride (fasudil, 30 mg/kg in 0.9% saline), MEK/ERK inhibitor (PD98059, 1 mg/kg in 5% DMSO, 5% Tween80, 80% PBS, Cayman Chemical) and PI3K inhibitor (Wortmannin, 0.25 mg/kg in 10% DMSO, 90% PBS, Selleck Chemical) were injected i.p.

## Western blot

10T1/2 RGS5myc cells were incubated in complete DMEM medium, or serum starved for 24 h followed by 24 h drug stimulation as indicated in figures. Cells were washed twice with PBS and lysed in radioimmunoprecipitation assay (RIPA, Sigma) buffer containing phenylmethylsulfonyl fluoride (PMSF), protease and phosphatase inhibitor cocktails (Sigma). Protein concentration was quantified using the bicinchoninic acid (BCA) assay kit (Thermo Fisher Scientific). A total of 20 μg protein was separated on a 12% SDS-PAGE gel and transferred onto a polyvinylidene difluoride (PVDF) membrane (Millipore). The membrane was incubated with blocking buffer (20 mM Tris, 150 mM NaCl, 0.1% Tween 20, 5% skim milk powder, pH 7.6) for 1 h at RT, followed by incubation with primary antibodies in blocking buffer overnight at 4 °C (primary antibodies and dilutions, Appendix Table S2). Membranes were incubated with horseradish peroxidase (HRP)-conjugated secondary antibodies (secondary antibodies, dilution 1:2000, Appendix Table S3). Signals were developed with enhanced chemiluminescence solution (Amersham) and visualized and quantified using the ChemiDoc™ MP Imaging System (Bio-Rad, software version 6.1).

## Mouse treatments

For 2-week treatment regimens, RIP1-Tag5 mice were treated from 25 to 27 weeks of age. For long-term treatments, 22-week-old RIP1-Tag5 mice were treated 5×/week for 2 weeks followed by 2×/week for 6 weeks, a total of 8 weeks. For orthotopic breast cancer induction, female C57BL/6 mice were injected with $5 \times 10^5$ AT3-OVA cells in 30 μl PBS in the fourth mammary fat pad. On day 8, mice were treated 5×/week for 3 weeks. For melanoma induction, $5 \times 10^5$ B16-OVA cells were injected intradermally (i.d.) into the flank of C57BL/6 mice. Mice were treated from day 5 to day 11. At end point, mice were anesthetized followed by transcardiac perfusion with 2% formalin. Prior to sacrifice, some mice were i.v. injected with 50 μg fluorescein isothiocyanate (FITC)-conjugated tomato lectin (Lycopersicon esculentum, Vector, circulated for 5 min) and pimonidazole i.p. (60 mg/kg, circulated for 60 min,

HypoxyprobeTM-1 Kit, Hypoxyprobe, Inc., USA). For adoptive transfer survival studies, mice were pre-treated with 5×/week i.v. injections of eribulin when tumors were palpable for a total of 2 weeks, followed by adoptive T cell transfers ($1 \times 10^6$ OT-I cells, i.v.) every third day from day 17 to 29 with one eribulin injection the day before adoptive T cell transfers. OVA-specific TCR transgenic T cells (OT-I) (Clarke et al, 2000) were in vitro activated with 10 U/ml IL-2 (Peprotech) and 25 nM OVA peptide 257–264 (SIINFEKL, Genscript) for 3 days. For checkpoint inhibitor survival studies, mice were pre-treated with 7 drug injections prior to 5× i.p. injections of 250 μg anti-PD1 antibodies (BioXCell) every third day from day 14 to 27 with one eribulin injection the day before each antibody injection. Tumor size was assessed by measurement of length and width with a microcaliper and was calculated using the formula (length × width$^2$)/2. The ethical endpoint was reached when tumors measured 1500 mm$^3$.

## Flow cytometry

Tumors were harvested in FACS buffer (1% FBS in PBS) and digested in 2.5 ml/0.1 g tumor tissue PBS containing 100 U/ml Collagenase IV, 0.5 mg/ml DNase I (both Worthington Biochemical). Erythrocytes were removed by 1 min incubation in ACK lysis buffer (0.15 M NH$_4$Cl, 10 mM KHCO$_3$, 0.1 mM EDTA in PBS). Cell suspensions were stained with ViaDye Violet Fixable Viability Dye Kit (CYTEK) for 15 min at room temperature for live cell detection. In total, $5 \times 10^6$ cells were blocked with Fc-block (CD16/CD32, clone 2.4G2, BioXCell) for 15 min on ice, and subsequently stained for 30 min on ice in FACS buffer with appropriate cell surface antibodies (Appendix Table S2). The true nuclear transcription factor buffer set (BioLegend) was used for all intracellular stainings following the manufacturer's protocol. After two washes in permeabilization buffer, cells were analyzed using the Aurora (Cytek) and Spectro Flo (Version 3.3.0, Cytek). For all samples, 50,000–500,000 live singlets were analyzed.

## Immunohistochemistry (IHC)

Following transcardiac perfusion with 2% formalin, tumors were isolated, post-fixed in 4% paraformaldehyde overnight and paraffin embedded, or incubated in 10% sucrose (2 h) followed by 30% sucrose overnight and frozen in OCT compound (Tissue Tek). Ice-cold acetone was used to fix 7-μm frozen sections before IHC with the exception of pAKT staining where frozen sections were post-fixed in 2% formalin. For patient samples, 5 μm paraffin sections were obtained from archival SOLTI-1007 breast cancer specimens. Paraffin sections were deparaffinized, rehydrated and quenched in

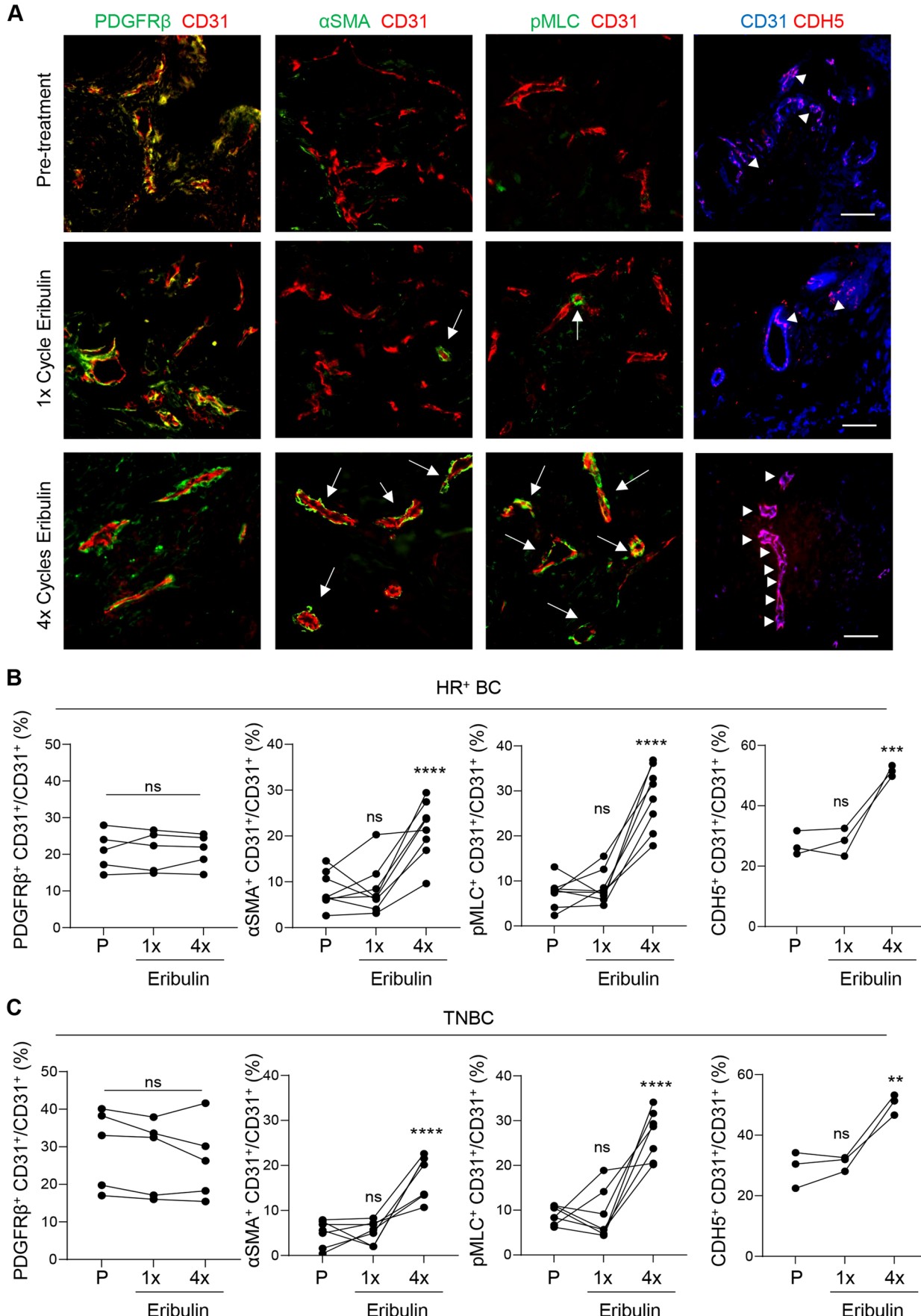

**Figure 8. Neoadjuvant eribulin treatment stabilizes blood vessels in human breast cancer.**

(A) Representative fluorescent micrographs show vascular markers in pre-treatment breast cancer biopsies, biopsies after one cycle of eribulin treatment and surgical specimens following four cycles of eribulin treatment. Tumor blood vessels (CD31, red) were assessed for overall PDGFRβ⁺ pericyte coverage (green), or vascular expression of the contractile markers αSMA (green) and pMLC (green). Arrows indicate αSMA⁺ or pMLC⁺ vessel coverage. VE-Cadherin expression (CDH5, red) on blood vessels (CD31, blue) is depicted as readout for endothelial barrier integrity. Arrow heads indicate VE-Cadherin expression on endothelial cells. Scale bars, 50 μm. (B) Individual patient data are shown. HR⁺ breast cancer patients: quantification of pericyte/endothelial cell coverage (PDGFRβ⁺ CD31⁺/CD31⁺ (%)) ($n = 5$, ns, not statistically significant), vascular αSMA ($n = 8$, ****$P < 0.0001$), pMLC ($n = 8$, ****$P < 0.0001$), and CDH5 expression ($n = 3$, ***$P = 0.0005$). (C) TNBC patients: quantification of pericyte/endothelial cell coverage ($n = 5$, ns), vascular αSMA ($n = 7$, ****$P < 0.0001$), pMLC ($n = 7$, ****$P < 0.0001$), and CDH5 expression ($n = 3$, **$P = 0.002$). Data were analyzed using one-way ANOVA. Patient data are summarized in Appendix Table S1. Source data are available online for this figure.

3% $H_2O_2$ in $H_2O$, followed by antigen retrieval. Primary antibodies are listed in Appendix Table S3. All primary antibodies were diluted 1:100 with the exception of NG2 (1E6.4) and pAKT (D9E) which were diluted 1:20 and 1:50, respectively. Primary antibodies were detected using the M.O.M. (Mouse on Mouse) immunodetection kit (Vector), tyramide kit (Thermo Fisher Scientific), fluorescein- (Vector) or streptavidin (SA)-conjugated AlexaFluor (AF) 594 (Thermo Fisher Scientific) and fluorescent-conjugated anti-IgG antibodies (secondary antibodies, dilution 1:1000, Appendix Table S3). 4',6-diamidino-2-phenylindole (DAPI) was used in some tissues to visualize cell nuclei. A Nikon Ti-E microscope and NIS software (Nikon, version 4.0) were used for image analysis. At least three mice or tumors were analyzed per treatment group; 3–15 images per tumor were analyzed. All material summarized in one graph was imaged with standardized threshold intensity. Positively stained features are expressed as % marker expression compared to total tumor surface area in one image (surface area %). Alternatively, co-localization was measured as fluorescence intensity ratio between red and green fluorescence channels or % overlay of red/green fluorescence.

## Bioinformatic analysis

Normalized gene expression data (microarray data) were downloaded from the Molecular Taxonomy of Breast Cancer International Consortium (METABRIC) using cBioPortal (https://www.cbioportal.org/study/summary?id=brca_metabric). Additional z-scoring was applied to standardize the expression data for survival analysis. Only patients with both survival information and expression data were included in this study. A total of 1483 patients (837 alive, 646 dead) were analyzed. The Cancer Genome Atlas (TCGA) gene expression data (RNAseq data) for breast cancer was obtained from the Genomic Data Commons (GDC, available at: https://portal.gdc.cancer.gov/) data portal. Data were normalized using EdgeR's Trimmed Mean of M-values (version 3.40.2) (Robinson and Oshlack, 2010). A total of 1083 samples (932 alive, 151 dead) were analyzed. Gene signatures for contractile and synthetic markers were generated from vascular markers identified in our cancer studies (Johansson-Percival et al, 2015; Li et al, 2024), and deduced from a metastasis-specific gene signature of primary tumors (Ramaswamy et al, 2003). The contractile gene signature is composed of the following genes: ACTG2, ACTA2, CNN1, CALD1, MYLK, MYH11, MYOCD, CDH5. The synthetic gene signature comprises: NOTCH3, RGS5, KLF4, LMNB1, COL1A1 (and COL1A2 for the TCGA data set). Overall survival was used as the primary prognosis endpoint. For the multivariant survival analysis, GSVA R package (version

### The paper explained

#### Problem

Blood vessels regulate drug and immune cell access into the tumor microenvironment. Destroying blood vessels in highly vascular, solid cancers—also known as anti-angiogenesis therapy—reduces tumor burden initially but also increases tumor hypoxia which fosters immune escape and metastatic dissemination. In contrast, vessel normalization alleviates hypoxia and improves immunotherapy. However, current approaches to normalize tumor blood vessels are transient, in particular when targeting the VEGF-pathway which is essential for endothelial cell survival. Thus, alternative approaches to achieve durable vessel remodeling and improve tumor oxygenation are required. Microtubule-binding drugs are widely used anti-cancer chemotherapeutics which can also modulate tumor stroma. Here, we investigated the mechanism of blood vessel remodeling, durability and consequences for immunotherapy.

#### Results

Comparison of 4 different microtubule-binding drugs revealed that combretastatin-A4 and eribulin act on angiogenic pericytes, the mural support cells of endothelial cells, in vitro and induce contractile markers in association with Rho kinase activation. In mouse cancer models in vivo, these drugs effectively switch pericyte phenotype from synthetic (or immature, proliferative) to contractile (or mature, more quiescent) which normalizes the entire vascular bed. This in turn reduces tumor hypoxia, changes the intratumoral immune landscape and facilitates immunotherapy. Importantly, extended treatments demonstrate that pericyte phenotype switching is highly durable in contrast to VEGF depletion. Moreover, pericyte phenotype switching was documented in human breast cancer patients treated with eribulin, and high expression levels of contractile markers correlate with better survival outcomes in breast cancer patients.

#### Impact

This study demonstrated that selective microtubule-binding drugs, in a dose-dependent manner, can be re-purposed to durably modify the cancer microenvironment and effectively "pre-condition" cancers for subsequent immunotherapy. Since drugs are already in clinical use, this study is of high translational impact.

1.46.0) was used to calculate an enrichment score using a Gaussian kernel function (suitable for continuous expression data) (Hanzelmann et al, 2013). Patients were stratified in low and high expression using a data-driven cutoff method (maximally selected rank statistics, maxstat R package version 0.7-25) (Hothorn and Zeileis, 2008). Survival outcomes of high and low expression signatures were compared by log-rank tests and plotted as Kaplan–Meier curves using the Survminer R package (version 0.4.9).

## Statistical analysis

GraphPad Prism software (versions 9 and 10) was used for statistical analyses. Data are presented as mean ± SEM or mean ± SD as indicated in figure legends. Sample sizes and $P$ values are shown in figure legends. No data were excluded from the analysis. For comparison of groups one- or two-way ANOVA with post hoc Tukey testing, or two-tailed unpaired Student's $t$ tests were used as indicated in figure legends. Survival data were analyzed using log rank (Mantel-Cox) tests. R version 4.2.2 was used for bioinformatic analyses. $P$ values <0.05 were considered significant.

# Data availability

This study includes no data deposited in external repositories.

The source data of this paper are collected in the following database record: biostudies:S-SCDT-10_1038-S44321-025-00222-6.

# Peer review information

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

## Acknowledgements

We acknowledge the support of the Harry Perkins Institute FACS facility. This work was funded by grants from the National Health and Medical Research Council (NHMRC, APP1141847), Worldwide Cancer Research (21-0257), Cancer Australia (PO411), Cancer Council Western Australia (1168) to RG and Cancer Australia (2002303) to BH. PKD is funded by NHMRC Investigator grant 2026403. The Translational Research Institute receives support from the Australian government. We thank Eisai Inc. for their financial support of the SOLTI-1007 trial and extend our gratitude to the patients who participated in the study.

## Author contributions

**Bo He**: Conceptualization; Data curation; Formal analysis; Supervision; Funding acquisition; Investigation; Visualization; Methodology; Writing—original draft; Writing—review and editing. **Kira H Wood**: Data curation; Formal analysis; Investigation; Visualization; Methodology; Writing—original draft; Writing—review and editing. **Zhi-jie Li**: Data curation; Formal analysis; Investigation; Methodology; Writing—original draft. **Judith A Ermer**: Formal analysis; Investigation; Visualization; Methodology. **Ji Li**: Investigation. **Edward R Bastow**: Investigation; Visualization; Methodology. **Suraj Sakaram**: statistical analysis of human breast cancer data sets. **Phillip K Darcy**: Resources; Writing—original draft. **Lisa J Spalding**: Project administration. **Cameron T Redfern**: Investigation. **Jordi Canes**: Writing—original draft; SOLTI data base and sample management coordination. **Mafalda Oliveira**: SOLTI-1007 recruiting site representative. **Aleix Prat**: SOLTI-1007 co-investigator, recruiting site. **Javier Cortes**: SOLTI-1007 principal investigator. **Erik W Thompson**: Resources; Writing—original draft. **Bruce A Littlefield**: Conceptualization; Resources; Funding acquisition; Writing—original draft. **Andrew Redfern**: Conceptualization; Resources; Writing—original draft; Project administration. **Ruth Ganss**: Conceptualization; Data curation; Formal analysis; Supervision; Funding acquisition; Visualization; Writing—original draft; Project administration; Writing—review and editing.

Source data underlying figure panels in this paper may have individual authorship assigned. Where available, figure panel/source data authorship is listed in the following database record: biostudies:S-SCDT-10_1038-S44321-025-00222-6.

## Disclosure and competing interests statement

PKD reports research funding from Myeloid Therapeutics, Prescient Therapeutics, Bristol-Myers-Squibb and Juno Therapeutics outside of this study. MO reports research support from AstraZeneca, Ayala Pharmaceuticals,

Boehringer-Ingelheim, Genentech, Gilead, GSK, Immutep, Roche, Seagen, Zenith Epigenetics; honoraria from AstraZeneca, Eisai, Gilead, Libbs, Lilly, MSD, Novartis, Pfizer, Roche, Seagen outside of this study. JC reports research support from Roche, Ariad Pharmaceuticals, AstraZeneca, Baxalta GMBH/ Servier Affaires, Bayer Healthcare, Eisai, F. Hoffman-La Roche, Guardanth Health, Merck Sharp&Dohme, Pfizer, Piqur Therapeutics, Iqvia; honoraria from Roche, Novartis, Eisai, Pfizer, Lilly, Merck Sharp&Dohme, Daiichi Sankyo, Astrazeneca, Gilead, Steamline Therapeutics outside of this study. BAL is an employee of Eisai Inc. which discovered, developed and now manufactures and markets the clinically formulated mesylate salt form of eribulin as Halaven®. AR had previous roles on advisory boards and speaker engagements for Eisai Inc. outside of this study. RG is a member of the Advisory Editorial Board of EMBO Molecular Medicine. This has no bearing on the editorial consideration of this article for publication. RG received financial support from Eisai Inc. for part of this research project, and provision of eribulin for this study. The remaining authors declare no competing interests.

# Expanded View Figures

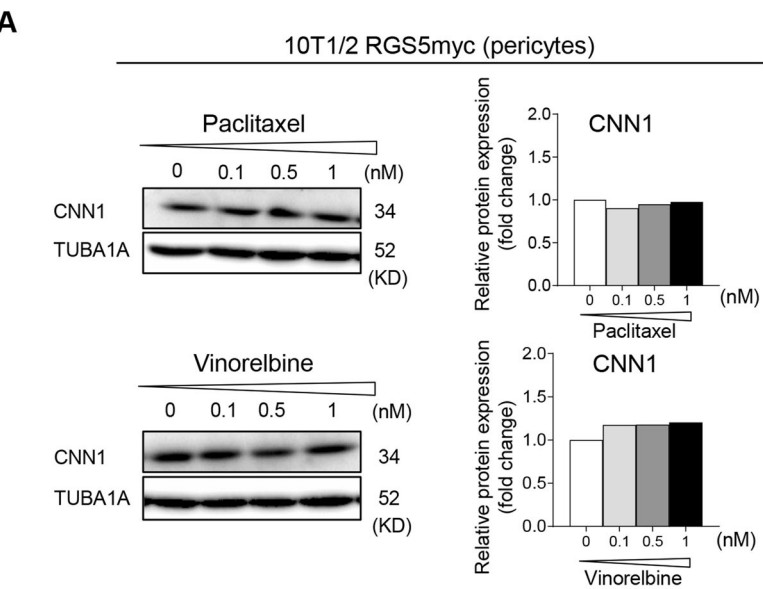

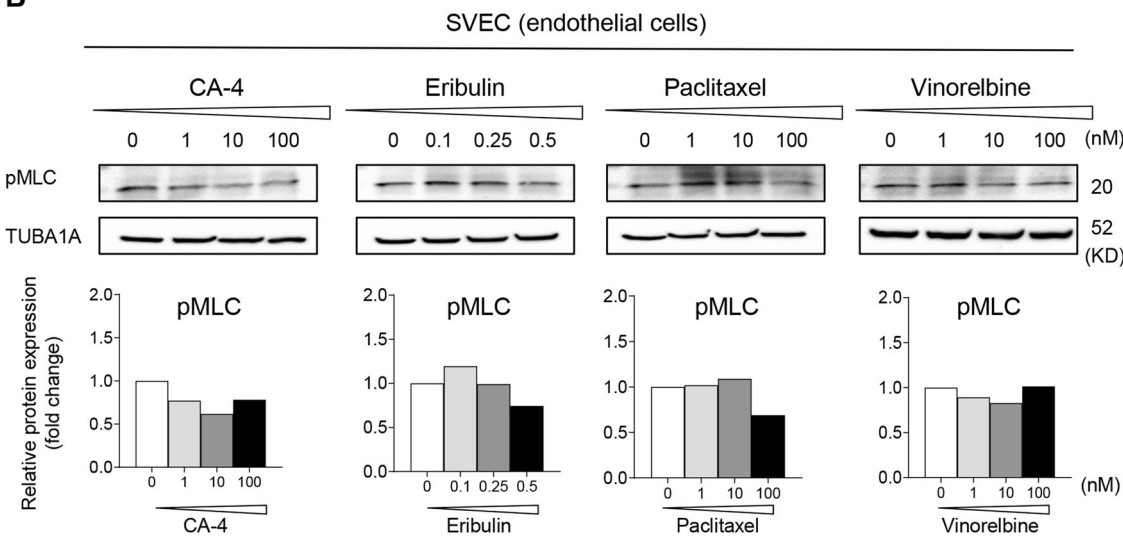

**Figure EV1. Low dose paclitaxel or vinorelbine do not induce pericyte maturity and all microtubule-binding drugs fail to activate RhoA kinase in endothelial cells.**

(A) Representative western blots from 10T1/2 RGS5myc cells incubated with indicated doses of paclitaxel or vinorelbine for 24 h. The contractile marker CNN1 intensities were quantified by densitometric analysis and normalized on tubulin intensity ($n = 1$). (B) Representative western blots from SVEC endothelial cells incubated with indicated doses of CA-4, eribulin, paclitaxel or vinorelbine for 24 h. pMLC intensities were quantified by densitometric analysis and normalized on tubulin intensity ($n = 1$).

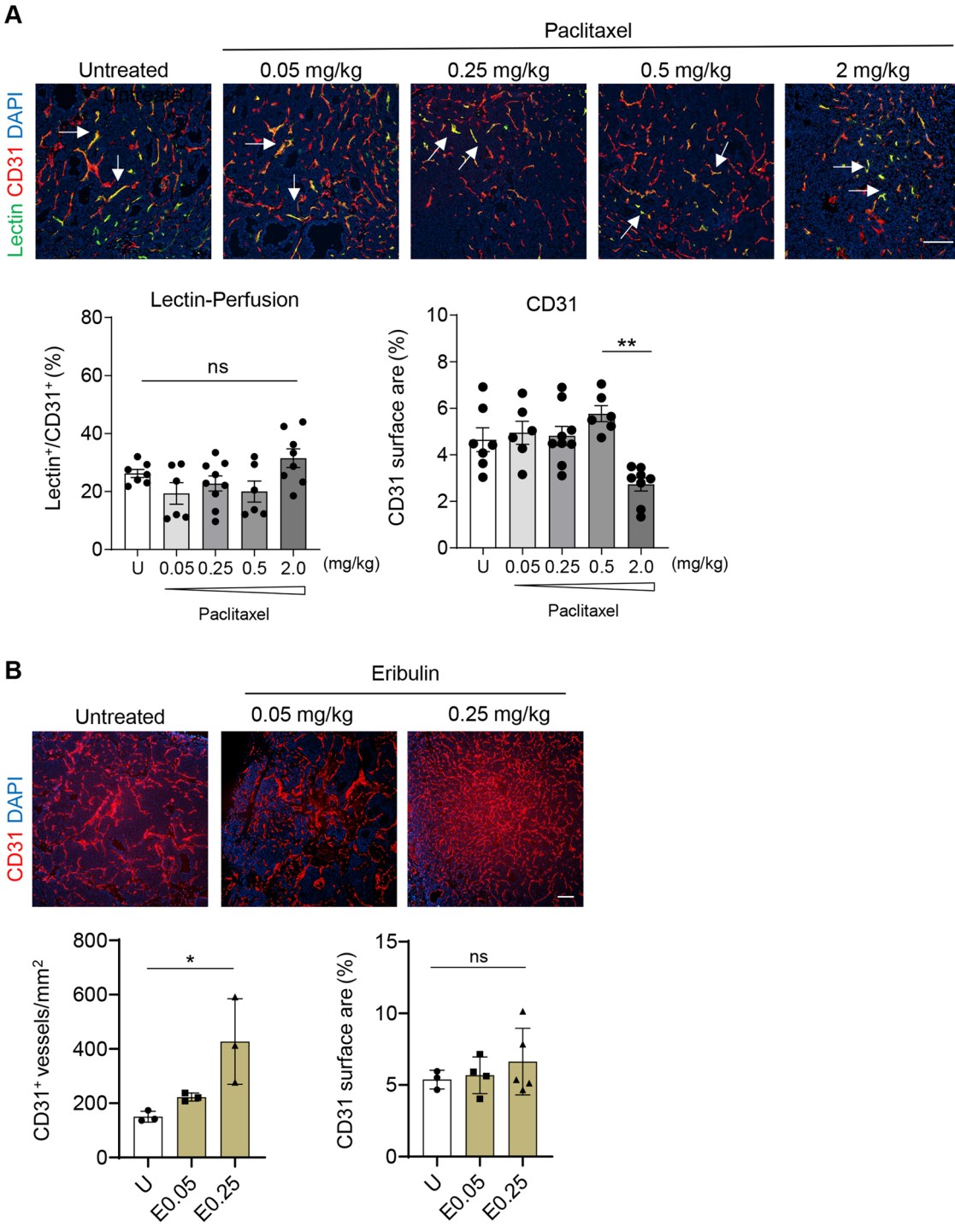

**Figure EV2. Metronomic paclitaxel dosing and eribulin have vessel remodeling effects.**

(A) Tumor-bearing RIP1-Tag5 mice were left untreated (U) or treated for 2 weeks with indicated doses of paclitaxel followed by i.v. infusion of FITC-lectin. Representative fluorescence micrographs showing FITC-lectin (green) overlay (yellow, arrows) with CD31 vascular (red) staining. Lectin perfusion and CD31 surface area (vascularity) were quantified, $n = 7$ tumors for untreated, $n = 6$ tumors for 50 mg/kg and 500 mg/kg, $n = 9$ tumors for 500 mg/kg and $n = 8$ tumors for 2 mg/kg paclitaxel. **$P = 0.0081$, ns, not statistically significant. Data were analyzed by one-way ANOVA and expressed as mean ± SEM. Scale bar, 100 μm. (B) Tumor-bearing RIP1-Tag5 mice were treated with 0.05 mg/kg (E0.05) or 0.25 mg/kg (E0.25) eribulin for 2 weeks and tumors stained with the vascular marker CD31 (red). Representative fluorescence micrographs show examples of tumor vascularity under eribulin treatment. Microvascular density (CD31+ vessels/mm², left, $n = 3$) and overall vascularity (CD31+ surface area, right, $n = 3$ for U, $n = 4$ for E0.05, $n = 5$ for E0.25) were quantified, *$P = 0.018$, E0.25 compared to untreated; ns, not statistically. Scale bar, 100 μm. Data were analyzed by one-way ANOVA and expressed as mean ± SEM.

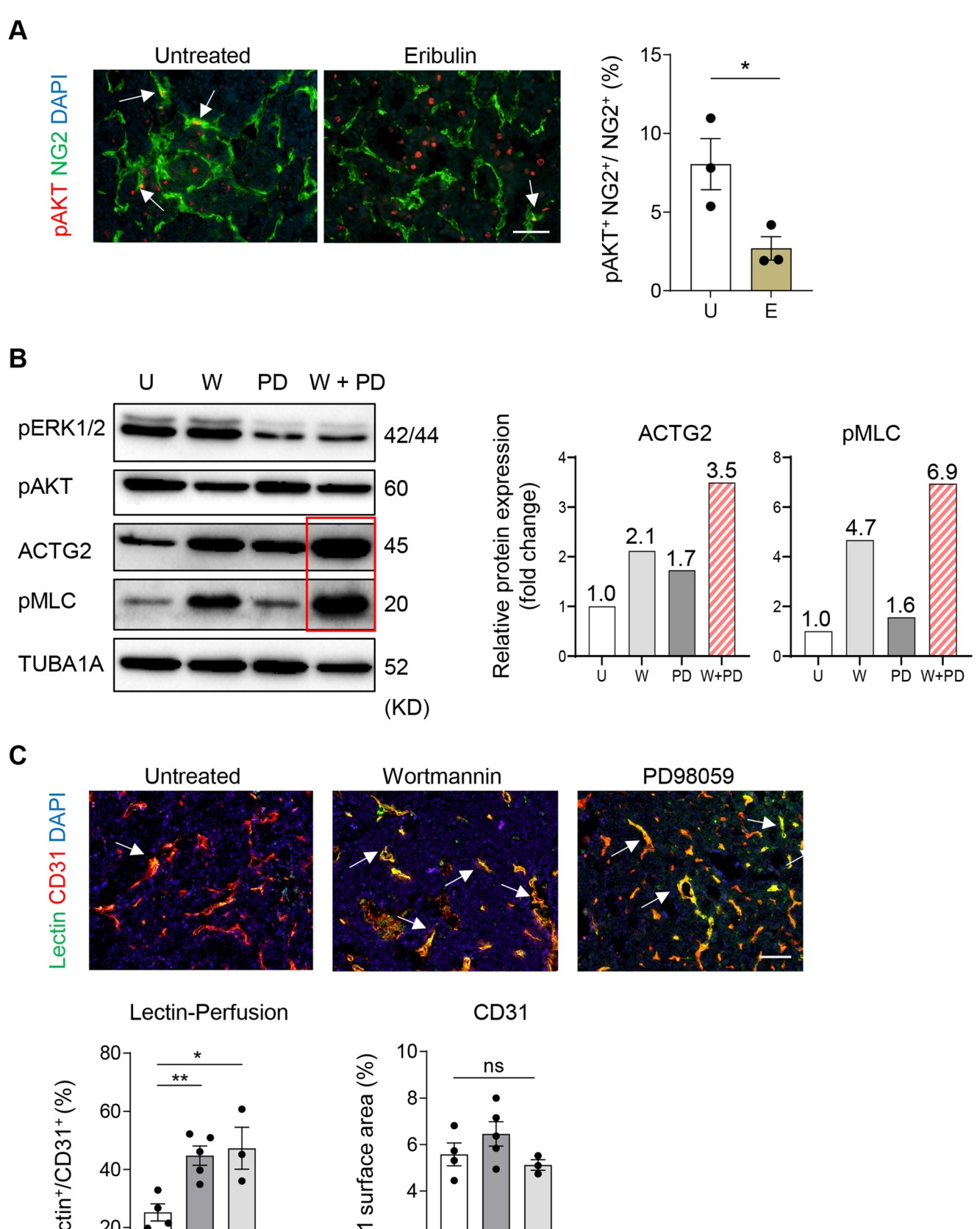

◀

**Figure EV3. Specific inhibition of ERK and/or PI3K signaling induces pericyte maturity and improves tumor perfusion.**

(A) Representative fluorescent micrographs show pAKT (red) expression in RIP1-Tag5 NG2$^+$ tumor pericytes (green). Arrows indicate overlay (yellow). Quantification of NG2$^+$ pericyte specific pAKT signals in untreated or eribulin treated tumors, $n = 3$ mice, $*P = 0.0402$. Data were analyzed using two-tailed, unpaired Student's $t$ test and presented as mean ± SEM. Scale bar, 50 μm. (B) Representative western blots from 10T1/2 RGS5myc cells incubated with signaling pathway inhibitors for 24 h (U, untreated; W, Wortmannin, 10 μM; PD, PD98058, 20 μM; W + PD, combination of Wortmannin and PD98058). Phosphorylated ERK1/2 and AKT proteins are shown as controls for PD98058 and Wortmannin activities, respectively. Phosphorylated MLC intensities were quantified by densitometric analysis and normalized to tubulin intensity to assess changes in activation; ACTG2 intensities were quantified to determine contractile maker induction ($n = 1$). Red boxing highlights additive marker induction following W + PD treatment. One of 2 independent experiments is shown. (C) Tumor-bearing RIP1-Tag5 mice were left untreated (U, $n = 4$) or treated 3 x/week for 2 weeks with Wortmannin (W, 0.25 mg/kg, $n = 5$) or PD98059 (PD, 1 mg/kg, $n = 3$) followed by i.v. infusion of FITC-lectin. Representative fluorescence micrographs showing FITC-lectin (green) overlay with CD31 vascular (red) staining as surrogate marker for tumor perfusion (yellow, arrows). Lectin perfusion and CD31 surface area (vascularity) were quantified, $*P = 0.0135$, $**P = 0.0130$, ns, not statistically significant. Data were analyzed by one-way ANOVA and expressed as mean ± SEM. Scale bar, 100 μm.

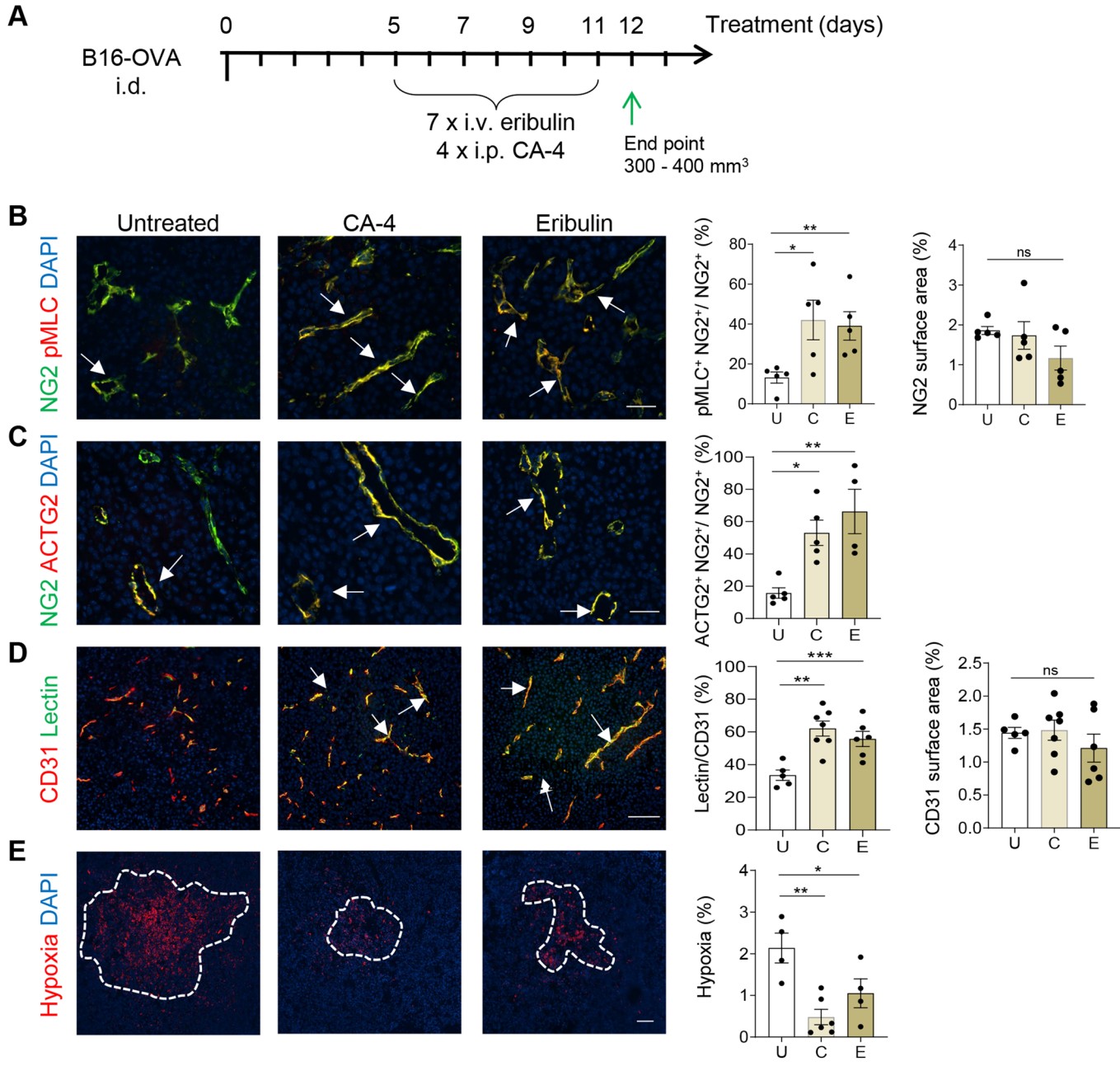

**Figure EV4. CA-4 and eribulin induce pericyte phenotype switching in mouse melanoma.**

(A) Treatment schedule of an orthotopic melanoma model (B16-OVA) and endpoint for immunohistochemistry. B16-OVA tumors were grown in untreated C57BL/6 mice (U), or mice treated with C-A4 (C, 2.5 mg/kg) or eribulin (E, 0.25 mg/kg). (B) Representative fluorescence micrographs show examples of drug-induced changes in pericyte pMLC expression. pMLC (red) coverage of NG2$^+$ pericytes (green) was quantified, arrows indicate overlay (yellow), $n = 5$, *$P = 0.048$, **$P = 0.028$, ns, not statistically significant. Scale bar, 50 μm. (C) Representative fluorescence micrographs and quantification of the contractile marker ACTG2 (red) expression in pericytes (NG2, green); arrows indicate overlay (yellow), $n = 5$ mice for U, C, $n = 4$ mice for E, *$P = 0.016$, **$P = 0.0034$. Scale bar, 50 μm. (D) Representative fluorescence micrographs and quantification of FITC-lectin (green) overlay (yellow, arrows) with CD31$^+$ (red) blood vessels as surrogate marker for tumor perfusion, $n = 5$ mice for U, $n = 7$ mice for C, $n = 6$ mice for E, **$P = 0.007$, ***$P = 0.0007$, ns, not statistically significant. Scale bar, 100 μm. (E) Representative fluorescence micrographs and quantification of hypoxyprobe (red, white dashed line) as marker for tumor oxygen status, $n = 4$ mice for U, E, $n = 6$ mice for C, *$P = 0.0477$, **$P = 0.0025$. Scale bar, 100 μm. All data were analyzed using one-way ANOVA and expressed as mean ± SEM.

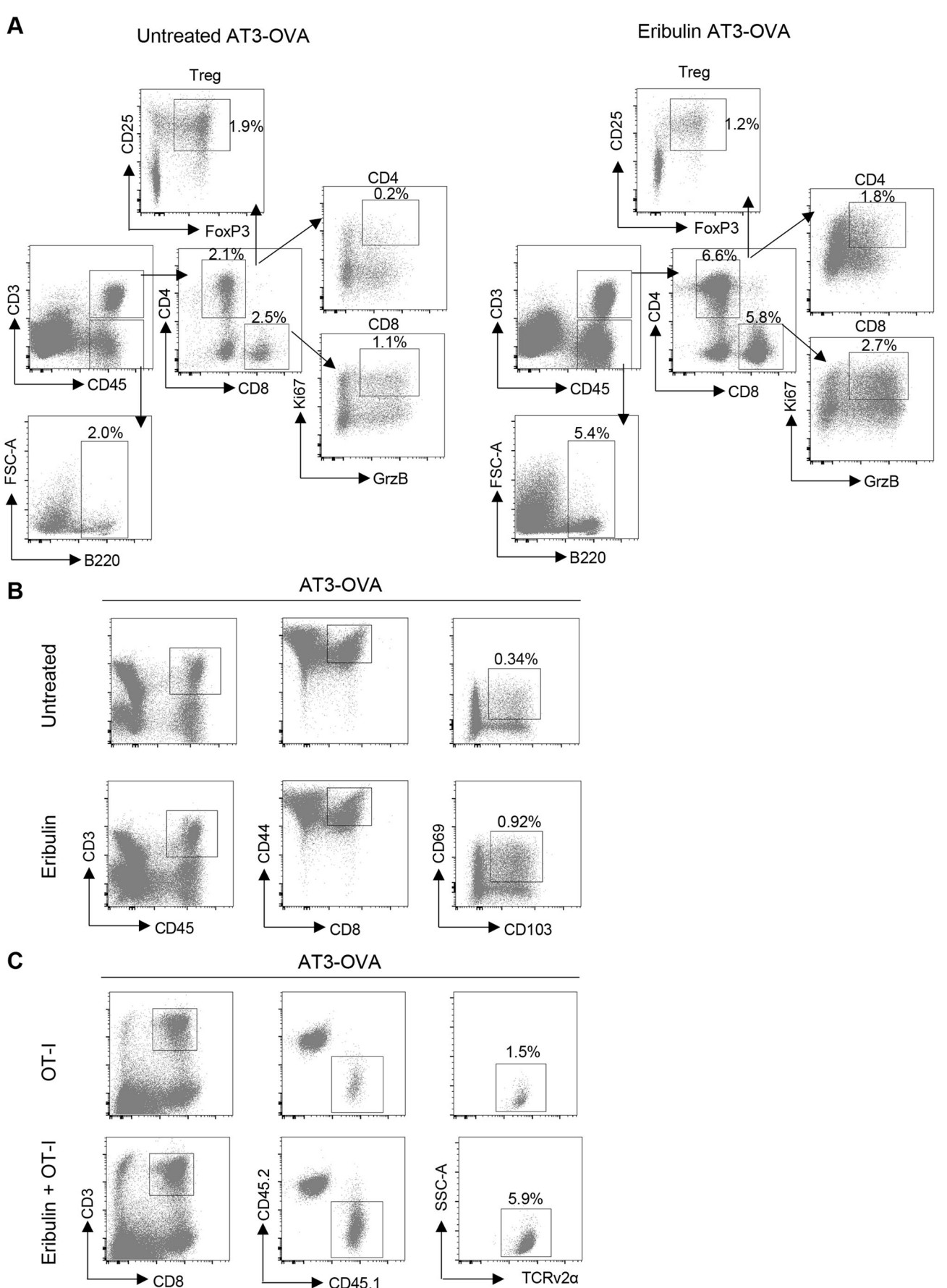

◀ **Figure EV5. Eribulin treatment increases spontaneous and adoptively transferred effector T cells in the breast cancer microenvironment.**

(A) Representative FACS blots showing gating strategy for FACS quantification of tumor-infiltrating CD4$^+$ T cells, effector CD4$^+$ T cells, T regs, CD8$^+$ T cells, effector CD8$^+$ T cells, and B cells in untreated (left) and eribulin treated AT3-OVA breast cancers (right). (B) Gating strategy for the detection of T$_{RM}$ CD8$^+$ T cells in untreated and eribulin treated AT3-OVA tumors. (C) Representative FACS blots showing gating strategy for FACS quantification of adoptively transferred OT-I T cells in untreated or eribulin treated AT3-OVA breast cancers.

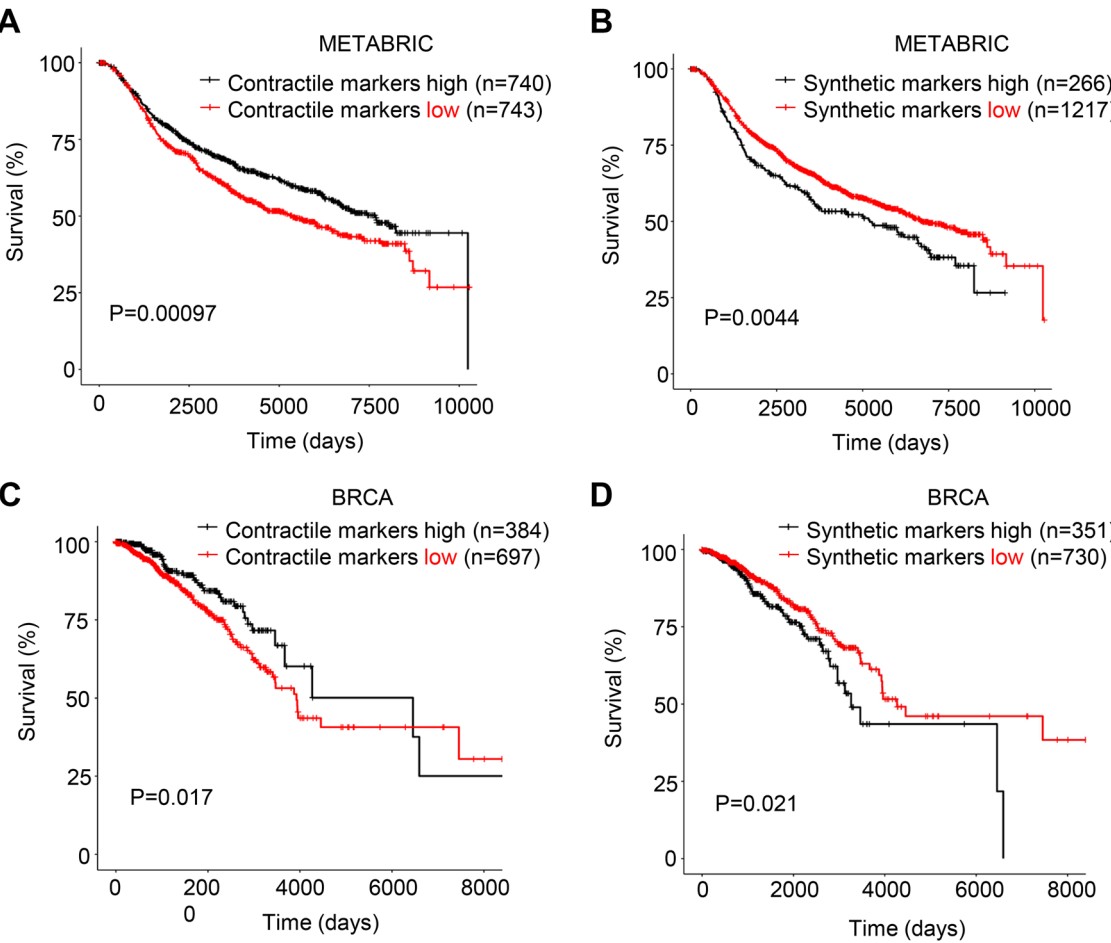

**Figure EV6. Improved survival outcome in breast cancer patients correlates with a contractile pericyte gene signature.**

(A, B) Kaplan–Meier curves showing prognostic value of (A) contractile and (B) synthetic pericyte gene signatures for disease progression in the METABRIC breast cancer patient cohort ($n = 1483$, 837 alive, 646 dead) using an optimal cutoff stratification to divide patients into high and low expression groups. Contractile gene signature: ACTG2, ACTA2, CNN1, CALD1, MYLK, MYH11, MYOCD, CDH5. Synthetic gene signature: NOTCH3, RGS5, KLF4, LMNB1, COL1A1, COL1A2. Log-rank test, (A) $P = 0.00097$; (B) $P = 0.0044$. (C, D) Kaplan–Meier curves showing prognostic value of (C) contractile and (D) synthetic pericyte gene signatures for disease progression in the BRCA breast cancer patient cohort ($n = 1083$, 932 alive, 151 dead) using an optimal cutoff stratification to divide patients into high and low expression groups. Contractile genes as above. Synthetic gene signature: NOTCH3, RGS5, KLF4, LMNB1, COL1A1. Log-rank test, (C) $P = 0.017$; (D), $P = 0.021$.

