## [Peer Review File · EMBO Molecular Medicine]

Selective tubulin-binding drugs induce pericyte phenotype switching and anti-cancer immunity

Bo He, Kira Wood, Zhi-jie Li, Judith Ermer, Ji Li, Edward Bastow, Suraj Sakaram, Phillip Darcy, Lisa Spalding, Cameron Redfern, Jordi Canes, Mafalda Oliveira, Aleix Prat, Javier Cortes, Erik Thompson, Bruce Littlefield, Andrew Redfern, and Ruth Ganss

Corresponding author(s): Ruth Ganss (ganss@perkins.org.au)

Review Timeline:

Submission Date:	10th Oct 24
Editorial Decision:	5th Nov 24
Revision Received:	5th Feb 25
Editorial Decision:	20th Feb 25
Revision Received:	23rd Feb 25
Accepted:	28th Feb 25

Editor: Lise Roth

Transaction Report:

5th Nov 2024

Dear Ruth,

Thank you for the submission of your manuscript to EMBO Molecular Medicine. We have now heard back from the referees who reviewed your manuscript. As you will see below, the reviewers raise substantial concerns on your work, which unfortunately preclude its publication in EMBO Molecular Medicine in its current form.

The reviewers find that the question addressed by the study is of potential translational interest, however referees #1 and #2 question the conceptual novelty of the findings, and referee #1 further raises concerns on rigor and reproducibility.

If you feel you can satisfactorily address these points and those listed by the referees, you may wish to submit a revised version of your manuscript. Please attach a covering letter giving details of the way in which you have handled each of the points raised by the referees. A revised manuscript will once again be subject to review, and we cannot guarantee at this stage that the eventual outcome will be favorable.

We are expecting your revised manuscript within three months, if you anticipate any delay, please contact us.

We require:

4) A .docx formatted letter INCLUDING the reviewers' reports and your detailed point-by-point responses to their comments. As part of the EMBO Press transparent editorial process, the point-by-point response is part of the Review Process File (RPF), which will be published alongside your paper.

5) A complete author checklist, which you can download from our author guidelines (<https://www.embopress.org/page/journal/17574684/authorguide#submissionofrevisions>). Please insert information in the checklist that is also reflected in the manuscript. The completed author checklist will also be part of the RPF.

6) All Materials and Methods need to be described in the main text using our 'Structured Methods' format. According to this format, the Methods section includes a Reagents and Tools Table (listing key reagents, experimental models, software and relevant equipment and including their sources and relevant identifiers) followed by a Methods and Protocols section describing the methods, ideally using a step-by-step protocol format. The aim is to facilitate adoption of the methodologies across labs. Please download and fill our Reagents and Tools Table template (.docx), which you can find in our author guidelines:

<https://www.embopress.org/doi/10.15252/msb.20178071>

7) It is mandatory to include a 'Data Availability' section after the Materials and Methods. Before submitting your revision, primary datasets produced in this study need to be deposited in an appropriate public database, and the accession numbers and database listed under 'Data Availability'. Please remember to provide a reviewer password if the datasets are not yet public (see <https://www.embopress.org/page/journal/17574684/authorguide#dataavailability>).

8) For data quantification: please specify the name of the statistical test used to generate error bars and P values, the number (n) of independent experiments (specify technical or biological replicates) underlying each data point and the test used to calculate p-values in each figure legend. The figure legends should contain a basic description of n, P and the test applied. Graphs must include a description of the bars and the error bars (s.d., s.e.m.). Please provide exact p values.

12) Author contributions: CRedit has replaced the traditional author contributions section because it offers a systematic machine readable author contributions format that allows for more effective research assessment. Please remove the Authors Contributions from the manuscript and use the free text boxes beneath each contributing author's name in our system to add specific details on the author's contribution. More information is available in our guide to authors.

13) Every published paper now includes a 'Synopsis' to further enhance discoverability. Synopses are displayed on the journal webpage and are freely accessible to all readers. They include a short stand first (maximum of 300 characters, including space) as well as 2-5 one-sentences bullet points that summarizes the paper. Please write the bullet points to summarize the key NEW findings. They should be designed to be complementary to the abstract - i.e. not repeat the same text. We encourage inclusion of key acronyms and quantitative information (maximum of 30 words / bullet point). Please use the passive voice. Please attach these in a separate file or send them by email, we will incorporate them accordingly.

Please also suggest a visual abstract to illustrate your article as a PNG file 550 px wide x 300-600 px high. A cropped portion of this image will serve as thumbnail for the table of content on our webpage.

14) As part of the EMBO Publications transparent editorial process initiative (see our Editorial at <http://embomolmed.embopress.org/content/2/9/329>), EMBO Molecular Medicine will publish online a Review Process File (RPF) to accompany accepted manuscripts.

In the event of acceptance, this file will be published in conjunction with your paper and will include the anonymous referee reports, your point-by-point response and all pertinent correspondence relating to the manuscript. Let us know whether you agree with the publication of the RPF and as here, if you want to remove or not any figures from it prior to publication. Please note that the Authors checklist will be published at the end of the RPF.

I look forward to receiving your revised manuscript.

Yours sincerely,

Lise Roth

***** Reviewer's comments *****

Referee #1 (Comments on Novelty/Model System for Author):

Prior literature that provides rational but also reduces novelty of the study is not adequately presented in the introduction.

Concerns for rigor and reproducibility - in vitro data do not have replicates presented and conclusions are made in the results between data that are not actually analyzed together.

The relevance of and rationale for the use of several different cancer models (pancreatic, melanoma, breast) for this study is not clearly presented.

Referee #1 (Remarks for Author):

A major limitation of the manuscript is the poor introduction that does not adequately set up the extensive literature already published on the effects of microtubule targeted chemotherapeutics (particularly eribulin and CA-4) on the tumor vasculature and immune infiltration and how those factors relate to the fact that eribulin demonstrates a survival advantage in patients whereas CA-4 has yet to demonstrate the therapeutic efficacy necessary for approval after 20 years of clinical evaluations. Most strikingly, Niwa et al, which demonstrated the ability of eribulin (as well as its liposomal formulation) to increase immune infiltration to promote synergy with immune checkpoint inhibitors in part through its vascular remodeling activities, is completely absent from the introduction and only briefly mentioned in the discussion as supporting the current findings. Ultimately, the fact that eribulin stabilizes tumor vasculature to improve immune infiltration is already well established in the literature and the current data suggest a mechanism based only on correlative data that is not surprising considering the published effects of microtubule destabilizers on the implicated pathways (e.g. ROCK activation) which is not adequately referenced).

Publication in this or any journal requires that the authors provide a clear summary of what new information the current manuscript is contributing to the literature as compared to the many studies already published on the role of eribulin and CA-4 on tumor vasculature.

There are also multiple specific concerns regarding statements made in the manuscript as well as the rigor and reproducibility of the data that are provided in more detail below.

1) The statement in the introduction that the antiangiogenic stromal effects of tubulin binding drugs are likely unrelated to their microtubule binding activity are not supported by the references included. This statement should either be removed or include references that actually support this statement.

2) The well-established ability of microtubule destabilizers like CA-4 and eribulin to promote RhoA kinase activation through a mechanism of release of the Rho exchange factor GEF-H1, which is also involved in its immunological effects is not referenced. (Chang et al, Mol Biol Cell, 2008; Heck et al, Mol Biol Cell, 2012; Kashyap et al, Cell Rep, 2019).

3) Figure 1 - Lack of indication of number of replicates or error bars for in vitro data in figure 1 demonstrate lack of rigor. Inconsistencies in the signal for baseline levels of expression for the different drug treatments are concerning. In particular there is a striking difference in the levels of pMLC for baseline vinorelbine treatment as compared to those seen at baseline for CA-4

treatment even though the levels of tubulin are similar. The lack of consistency in baseline measures and lack of replicates make these data impossible to interpret in any meaningful way.

4) Figure 2 - dose of paclitaxel chosen was considered 'too high', however it is not evident why lower doses were not considered. Given the lack of investigating multiple doses, the statements that paclitaxel and vinorelbine had no effect on pericyte phenotypes should be tempered to indicate that no effect was observed at the selected concentrations.

5) Figure 5 - It is well-known that eribulin is more potent than taxanes, vinca alkaloids, and CA-4 due to its preferential poisoning at the plus end of microtubules. Therefore, in particular, the comparison of 1.5 mg/kg CA-4 to 0.25 mg/kg eribulin, vinorelbine, and paclitaxel in Figure 5 is not well rationalized and it is not surprising that these low doses of paclitaxel and vinorelbine have no observable effect. Overall concern that the lack of phenotypes at single doses are being interpreted as a mechanistic difference as opposed to a difference in potency.

6) Figure 6 - lack of eribulin only control limits conclusions that can be made. If data in Supplemental figure S4 on untreated and eribulin only tumors are to be compared to the data in Figure 6 with immunotherapy +/- eribulin as suggested in the results section (bottom of page 9), all data should be presented together in one graph. Increased CD4 infiltration by eribulin similar to that reported in other papers not referenced (Takahashi, *Cancers*, 2022; Keenan *nature communication*, 2021)

7) Figure 7 - Again, the results describe eribulin + OT-I treatment providing a significant survival advantage as compared to each treatment alone by comparing 7C and 7D, but these data are graphed separately on two different graphs with distinct statistical comparisons such that the single treatment eribulin is not directly compared to the combination and neither of the regimens with OT-I are compared to untreated or single treatment of eribulin. Data should be compared together in a single graph or the conclusions should be modified to only compare conditions that were directly evaluated together. Leads to concerns about reproducibility (why are untreated tumors in 7D growing slower than OT-I treated animals in 7C) and transparency of the data presented (statement of comparisons in the results that are not actually being made in the data analysis).

8) No rationale provided for the various models used. Pancreatic neuroendocrine tumors, ovalbumin expressing TNBC and melanoma models, and human samples from HER2 negative cancers and rationale for dosing differences between drugs and between experiments not provided.

9) Abbreviation of combretastatin A4 is not consistent with the literature. Should be CA-4, not C-A4

10) No data presented to support the overall conclusions that the pharmacological effects observed are mediated through ROCK signaling as all data are correlative. In the absence of this direct data, authors are advised to include references in the introduction describing the well-known effects of microtubule destabilizers on this pathway to provide additional rationale for the approach and conclusions.

11) The authors need to more rigorously address the vascular stabilizing effects of CA-4 in their models/dosing regimens as compared to the vascular disrupting effects ascribed to this compound in the broader literature

Referee #2 (Comments on Novelty/Model System for Author):

The role of pericyte maturation in anti-tumor immune response has been extensively studied. In the absence of novel mechanistic insights into the process of pericyte maturation the study appears to be confirmatory.

Referee #2 (Remarks for Author):

The study by He et al. reports that the tubulin-binding drugs combretastatin A4 (C-A4) and eribulin induce tumor-associated pericyte maturation, enhance tumor perfusion, and improve responses to adoptive transfer or immune checkpoint blockade (ICB) therapy in animal models. The authors propose that C-A4 and eribulin achieve these effects by activating RhoA signaling and suppressing the ERK1/2 and AKT pathways, promoting a switch from the synthetic to the contractile phenotype of pericytes. They further demonstrate that contractile pericytes are enriched in human tumors following eribulin treatment, with a contractile signature associated with better treatment outcomes.

Overall, this is a technically solid and well-written paper that effectively delivers its message. While the concept of vascular, and specifically pericyte, maturation and its synergy with immunotherapy is not entirely novel, the novelty here lies in the specific effects of two tubulin-binding drugs on pericyte phenotypes. The demonstration of similar effects in human patient samples enhances the translational relevance of these findings. I have the following specific comments:

1. Do C-A4 and eribulin induce the same effects-such as pMLCK activation and pERK1/2 and mTOR inhibition-in angiogenic endothelial cells both in vitro and in vivo, and to what extent does this contribute to the tumor phenotypes observed in vivo?
2. What is the proposed mechanism by which C-A4 and eribulin inhibit the synthetic phenotype? Specifically, how do these drugs activate RhoA, suppress pERK1/2 and the mTOR pathway, and induce the expression of contractile genes? Also, what is the impact of drugs on pericyte and endothelial cell proliferation?

3. The statement "eribulin induced contractile pericyte properties" lacks a functional readout to support this claim.
4. The statement "Mechanistically, C-A4 and eribulin both suppressed pERK signaling specifically in tumor pericytes (Fig. 3A)" lacks sufficient specificity due to the absence of endothelial cell analysis.
5. To substantiate the statement, "we demonstrated in mouse tumor models the selective capacity of low-dose C-A4 or eribulin to induce pericyte phenotype switching from a highly proliferative to a contractile state by activating RhoA kinase," the authors would need to either activate RhoA kinase in pericytes or show that inhibiting RhoA in pericytes prevents maturation in response to C-A4 and eribulin.

Technical comments:

1. Figure 1 shows increases in Cnn1, pMLCK, and ACTG2 in vitro upon treatment with C-A4 and eribulin, but not with other drugs. While the western blots are convincing, quantification is missing, and it is unclear how many times the experiment was repeated.
2. pS6R is a downstream target of the mTOR pathway. Although the AKT and mTOR pathways are mechanistically linked, they are not equivalent; using pAKT staining would provide a more accurate readout if the authors wish to emphasize AKT involvement.

Minor comment: In "vascular pericytes," remove "vascular."

Referee #3 (Comments on Novelty/Model System for Author):

This paper reports an interesting finding that has a potential for a high translational impact. The study is technically sound, I see no obvious issues.

Referee #3 (Remarks for Author):

The manuscript presents a finding that low concentrations of microtubule-binding drugs C-A4 and eribulin induce pericyte differentiation by inducing RhoA activation, leading to vasculature normalization. In turn, vasculature normalization reduced hypoxia, linked with more aggressive phenotypes of tumor cells, and enhanced the accessibility and effects of immune therapies.

The data linking tubulin-binding drugs with improved vascularization and enhanced therapeutic outcomes is compelling. The manuscript is well-written, and the data is logically and clearly presented. The reported findings have a high translational potential.

My only non-critical concern (as it does not really impact the translational potential of the reported findings) is about the inferences of the mechanisms, which is postulated to involve RhoA activation caused by simultaneous suppression of the MEK/ERK and AKT/PI3K. While plausible, the manuscript misses an obvious opportunity to validate their mechanistic inferences using RhoA inhibitor to suppress the effect of epirubicin and CA4, or a combination of MEK/AKT inhibitors to phenocopy the effect. Achieving the expected effect would strongly strengthen the mechanistic inferences, while the inability to achieve the effects would indicate the involvement of other or additional mechanisms, which would not compromise the impact of the study.

- Another obvious question is how microtubule binding drugs impact the observed signaling changes, and what could be the mechanisms responsible for the pericyte specificity. While addressing these questions might be outside of the scope of the paper, I suggest at least speculating on it in the discussion or explicitly mentioning the lack of knowledge about the mechanisms underlying the connection.

- While not essential, it would be highly desirable to validate vasculature normalization with an alternative vasculature-specific stain such as CD34 or MECA32, especially for the clinical samples where experimental approaches to characterize vasculature integrity and hypoxia are not accessible. That is if the authors have access to additional tissue sections from the trial.

- Finally, it would be desirable to provide evidence of RhoA activation and vasculature normalization for the B16-OVA model used in the adoptive T cell transfer study reported in Figure 7.

Minor points:

- A scale bar needs to be provided for the Figure 8A data.
- Page 7 "inducing higher and more coherent expression of ...VE-cadherin" - what does "more coherent expression" mean?

Consider explaining or rephrasing.

- Consider providing volumetric traces for individual tumors in the supplementary information.
- In the Results section, the authors refer to a non-existing panel 6H instead of the 6G, this needs to be fixed.

We thank the referees for their thoughtful comments and have substantially revised our manuscript accordingly.

Text highlighted in blue indicates changes to the manuscript text.

Referee #1 (Comments on Novelty/Model System for Author):

Prior literature that provides rationale but also reduces novelty of the study is not adequately presented in the introduction.

Concerns for rigor and reproducibility - in vitro data do not have replicates presented and conclusions are made in the results between data that are not actually analyzed together.

The relevance of and rationale for the use of several different cancer models (pancreatic, melanoma, breast) for this study is not clearly presented.

Our responses to these statements are provided below.

Referee #1:

A major limitation of the manuscript is the poor introduction that does not adequately set up the extensive literature already published on the effects of microtubule targeted chemotherapeutics (particularly eribulin and CA-4) on the tumor vasculature and immune infiltration and how those factors relate to the fact that eribulin demonstrates a survival advantage in patients whereas CA-4 has yet to demonstrate the therapeutic efficacy necessary for approval after 20 years of clinical evaluations. Most strikingly, Niwa et al, which demonstrated the ability of eribulin (as well as its liposomal formulation) to increase immune infiltration to promote synergy with immune checkpoint inhibitors in part through its vascular remodeling activities, is completely absent from the introduction and only briefly mentioned in the discussion as supporting the current findings. Ultimately, the fact that eribulin stabilizes tumor vasculature to improve immune infiltration is already well established in the literature and the current data suggest a mechanism based only on correlative data that is not surprising considering the published effects of microtubule destabilizers on the implicated pathways (e.g. ROCK activation) which is not adequately referenced).

Publication in this or any journal requires that the authors provide a clear summary of what new information the current manuscript is contributing to the literature as compared to the many studies already published on the role of eribulin and CA-4 on tumor vasculature.

Introduction:

To improve the introduction, we have expanded on the known biological effects of combretastatin and its limitations in current clinical development (Introduction, page 4/5):

“However, vascular disrupting agents have disappointing single agent activity due to neuro- and cardiovascular toxicity (Bates & Eastman, 2017) which opens the field to alternative applications.”

We also included a statement which makes it clear that eribulin has stromal/vascular modulating effects which improve immunotherapy, and cite Niwa et al. in the introduction as one of the publications which have shown synergistic effects (Introduction, page 5).

“Eribulin has shown synergistic effects with immunotherapy in mouse and human cancers suggesting potential vessel normalization mechanisms (Niwa et al, 2023; Tolaney et al, 2021).”

This statement does not impact on the novelty of our manuscript since the key question of our work is how eribulin (and combretastatin) stabilize angiogenic blood vessels.

Novelty:

The novelty of this manuscript is the discovery that eribulin and combretastatin induce pericyte phenotype switching in the angiogenic vasculature. Pericyte maturity and contractile marker expression are regulated through opposing roles of Rho kinase and ERK/AKT signaling.

To make this clearer to the reader we have revised the Introduction (page 5) as follows:

“Here we demonstrate that eribulin and combretastatin, but not paclitaxel or vinorelbine, induce pericyte phenotype switching in the angiogenic vasculature. Pericyte maturity and contractile marker expression are regulated through opposing roles of Rho kinase and ERK signaling.

Importantly, low dose treatment regimens improve overall tumor perfusion without killing cancer cells but act synergistically with immunotherapy. The application of established drugs to achieve novel therapeutic effects such as stromal remodeling creates highly translatable opportunities for therapeutic targeting, in particular for cancers which have been intrinsically resistant to current immunotherapy regimens.”

The novel translational implication is that pericyte maturity via phenotype switching stabilizes the angiogenic vasculature, increases tumor perfusion and reduces tumor hypoxia. It is important, but not surprising that this in turn enhances anti-tumor immunotherapy. We also demonstrate for the first time that pericyte phenotype switching can be induced in human breast cancer following neoadjuvant eribulin treatment. Importantly, this breast cancer cohort is a unique resource and the only eribulin neoadjuvant clinical trial with repeat biopsies.

The ability to regulate pericyte contractile phenotype by re-dosing/re-scheduling microtubule-binding therapeutics is novel and of high translation potential.

More specifically:

Combretastatin has only been investigated as a vascular disrupting agent in cancer; a low dose application to stabilize vessels remains unexplored.

Eribulin has indeed been described to modulate angiogenic blood vessels but it was unknown how. Besides delineating pericyte phenotype switching, we also demonstrate that targeting pericytes is highly durable compared to directly targeting/destroying endothelial cells. So far, there has been no clear differentiation between vascular effects of low dose anti-VEGF and eribulin.

Rho kinase (ROCK) activation in the vasculature has only been described for combretastatin (Williams et al 2014) which prompted us to investigate microtubule-binding drugs on pericytes (as described in the Introduction). The focus of this manuscript is on pericyte Rho kinase activity.

Importantly, the focus on pericytes is directly related to our earlier publication (Li et al., JCI 2024) which shows in genetic studies that activation of Rho kinase specifically in tumor pericytes (and no other cell type in the tumor microenvironment) normalizes the vascular bed and changes the microenvironment without directly affecting immune cells or cancer cells.

Whilst drug effects are not cell-type specific (compared to gene manipulations), drug use in this manuscript was titrated to meet 3 essential criteria:

- i. Pericyte Rho kinase activity and contractile markers were upregulated (as measured *in vitro* and *in vivo*).
- ii. Blood vessels were not destroyed (as measured by vessel frequency and functionally by tumor perfusion).
- iii. Cancer cell growth and Rho kinase activity remained unaffected (as measured by tumor growth, pERK/pS6R and pMLC expression *in vivo*).

That Rho kinase is upregulated by microtubule destabilization in epithelial breast cells, in HeLa cells, and in dendritic cells via GEF-H1 signaling is interesting. However, none of these studies describes signaling effects in pericytes (or vSMC) where Rho kinase upregulation leads to suppression of ERK/AKT signaling.

There are also multiple specific concerns regarding statements made in the manuscript as well as the rigor and reproducibility of the data that are provided in more detail below.

1) The statement in the introduction that the antiangiogenic stromal effects of tubulin binding drugs are likely unrelated to their microtubule binding activity are not supported by the references included. This statement should either be removed or include references that actually support this statement.

This statement had a broader intention than describing intracellular signaling following metronomic chemotherapy. It was related to potential anti-angiogenic effects involving for instance circulating endothelial progenitor cells. The concept is nicely summarized in Figure 1 in the cited paper (Scharovsky et al, 2009).

We have changed the statement to make this clearer to the reader (Introduction, page 5):

“These anti-angiogenic stromal effects are multi-faceted and may be unrelated to their anti-mitotic activity (Scharovsky et al, 2009; Wang et al, 2003).”

2) The well-established ability of microtubule destabilizers like CA-4 and eribulin to promote RhoA kinase activation through a mechanism of release of the Rho exchange factor GEF-H1, which is also involved in its immunological effects is not referenced. (Chang et al, Mol Biol Cell, 2008; Heck et al, Mol Biol Cell, 2012; Kashyap et al, Cell Rep, 2019).

These are very interesting studies and explain effector mechanisms based on microtubule destabilization by drugs such as nocodazole and eribulin (not combretastatin) in cancer/epithelial cells and immune cells. The results are different to our observations in angiogenic pericytes. For instance, microtubule destabilisation in dendritic cells (Kashyap et al, Cell Rep, 2019) is pro-inflammatory and associated with upregulation of ERK signaling. Response of breast epithelial cells to biomechanical changes via upregulation of GEF-H1 is independent of pERK (Heck, MBoC 2012). In contrast, eribulin/combretastatin treatment in pericytes leads to Rho kinase activation but downregulation of pERK.

3) *Figure 1 - Lack of indication of number of replicates or error bars for in vitro data in figure 1 demonstrate lack of rigor. Inconsistencies in the signal for baseline levels of expression for the different drug treatments are concerning. In particular there is a striking difference in the levels of pMLC for baseline vinorelbine treatment as compared to those seen at baseline for CA-4 treatment even though the levels of tubulin are similar. The lack of consistency in baseline measures and lack of replicates make these data impossible to interpret in any meaningful way.*

To address the Reviewer's concerns we now show in **revised Figure 1 B-E** quantifications of 3 independent experiments. As before, these combined data demonstrate that eribulin and combretastatin, but not paclitaxel or vinorelbine, activate RhoA kinase and enhance contractile marker expression.

We also replaced pMLC data generated from combretastatin (revised Figure 1B), paclitaxel (revised Figure 1D) and vinorelbine (revised Figure 1E) treated cells with alternative data sets or reduced exposure time. The results remain the same.

4) *Figure 2 - dose of paclitaxel chosen was considered 'too high', however it is not evident why lower doses were not considered. Given the lack of investigating multiple doses, the statements that paclitaxel and vinorelbine had no effect on pericyte phenotypes should be tempered to indicate that no effect was observed at the selected concentrations.*

The Reviewer is correct to point out that the conclusion shown in Figure 2 can only be made for the selected drug dose. We acknowledge this now in the revised manuscript text (Results, page 7):

“Paclitaxel and vinorelbine had no effect on pericyte phenotype *in vivo* at the applied doses (Fig. 2B-D),”...

However, the dose selection in Figure 2 is based on an extensive analysis of a range of doses *in vitro* and *in vivo* for their ability to induce pericyte contractile markers.

Revised Efigure EV1A now shows additional *in vitro* dosing of paclitaxel (and vinorelbine) covering lower concentrations (0/0.1/0.5/1/5 nM) than shown in Figure 1 (0/1/10/100 nM). We therefore demonstrate that paclitaxel and vinorelbine neither at a doses similar to combretastatin nor to eribulin induce pericyte phenotype switching.

Revised Efigure EV2A shows a comprehensive *in vivo* dosing range for paclitaxel from 0.05 mg/kg, 0.25 mg/kg, 0.5 mg/kg, and 2 mg/kg.

Please note that 2 mg/kg of paclitaxel induced vessel death *in vivo* and was therefore excluded from our study.

Importantly, neither high nor low dose ranges induced pericyte phenotype change and consequently enhanced tumor perfusion.

5) Figure 5 - It is well-known that eribulin is more potent than taxanes, vinca alkaloids, and CA-4 due to its preferential poisoning at the plus end of microtubules. Therefore, in particular, the comparison of 1.5 mg/kg CA-4 to 0.25 mg/kg eribulin, vinorelbine, and paclitaxel in Figure 5 is not well rationalized and it is not surprising that these low doses of paclitaxel and vinorelbine have no observable effect. Overall concern that the lack of phenotypes at single doses are being interpreted as a mechanistic difference as opposed to a difference in potency.

Drug applications in our manuscript are based on the potency of drug doses to induce vessel stabilization versus vessel death.

Specifically, paclitaxel/vinorelbine did not induce pericyte maturity at a viable dose range (e.g, a dose which did not induce vessel death), which differentiates them from eribulin/combretastatin. As such, we ruled out vessel-stabilizing effects of paclitaxel/vinorelbine but did not draw any mechanistic conclusions.

The rationale for using the drug doses in AT3-OVA breast cancer remain the same as delineated for RIP1-Tag5 tumors (Figure 2). To make this clearer we added the following sentence (Results, page 9)

“Drug dosing in AT3-OVA tumors was chosen to not affect tumor growth or reduce vessel counts.”

As discussed above, drug dosing was empirically determined for a therapeutic window where the vasculature is stabilized without inducing vessel death. Our *in vitro* data cover a wide range of drug concentrations which were used as pre-screen for *in vivo* experiments which confirm *in vitro* data (Figure 1 and **revised Figure EV2**). It is important to note, that drugs which induce pericyte maturity work over a range of concentrations, e.g. *in vitro* combretastatin induces pMLC and contractile markers from 10 nM to 1 mM. The effects we describe here are therefore not just applicable for a narrow dose range which could be easily missed.

Please also note that this application (vessel stabilization) is fundamentally different to the current use of microtubule-binding drugs. Independent of underlying mechanisms, paclitaxel and vinorelbine induced vessel death at doses where combretastatin still acted as a stabilizer.

Therefore, we were unable to compare paclitaxel/vinorelbine at the same effector dose as combretastatin (1.5 – 2.5 mg/kg).

6) *Figure 6 - lack of eribulin only control limits conclusions that can be made. If data in Supplemental figure S4 on untreated and eribulin only tumors are to be compared to the data in Figure 6 with immunotherapy +/- eribulin as suggested in the results section (bottom of page 9), all data should be presented together in one graph. Increased CD4 infiltration by eribulin similar to that reported in other papers not referenced (Takahashi, Cancers, 2022; Keenan nature communication, 2021)*

Addressing Reviewer 1's concern we have **revised Figure 6F** to show all treatment groups from the same experiment in one graph. The interpretation of our data remains the same.

We report a significant increase in CD4⁺, but also CD8⁺ T cell numbers and activation status. We have now added the suggested reference (Takahashi et al, 2022) which reports an increase in activated CD4⁺ T cells in spleens of breast cancer bearing mice following eribulin treatment. We have also added Oya et al, 2023, to acknowledge that eribulin treatment has been shown to correlate with increased T cell effector numbers (CD4⁺ and CD8⁺) which is dependent on CD103.

Results (page 10):

“In particular, total CD4⁺ T cell numbers increased concomitantly with a major shift in Ki67⁺ GrzB⁺ effector CD4⁺ T cells and reduced T reg cells (Fig. 6B, Fig. EV5A), consistent with an increase in activated CD4⁺ T cells in spleens of breast cancer-bearing mice following eribulin treatment (Takahashi et al, 2022)“...

...;T_{RM} have been associated with improved clinical outcome in TNBC patients and CD103⁺ tumor-infiltrating lymphocytes are essential for eribulin-mediated anti-tumor effects in mice (Oya et al, 2023; Virassamy et al, 2023).”

Suggested reference Keenan et al, 2011, found a positive correlation between responder groups of an eribulin + pembrolizumab HR⁺ breast cancer trial and overall lymphocyte infiltration, with more resting CD4⁺ T memory cells and follicular helper T cells in responder groups. In this particular study, it is difficult to distinguish between eribulin and pembrolizumab effects since this study compared responders and non-responders in double treatment.

7) *Figure 7 - Again, the results describe eribulin + OT-I treatment providing a significant*

survival advantage as compared to each treatment alone by comparing 7C and 7D, but these data are graphed separately on two different graphs with distinct statistical comparisons such that the single treatment eribulin is not directly compared to the combination and neither of the regimens with OT-I are compared to untreated or single treatment of eribulin. Data should be compared together in a single graph or the conclusions should be modified to only compare conditions that were directly evaluated together. Leads to concerns about reproducibility (why are untreated tumors in 7D growing slower than OT-I treated animals in 7C) and transparency of the data presented (statement of comparisons in the results that are not actually being made in the data analysis).

We have **revised Figure 7C** to show all treatment groups from the same experiment in one graph. As stated in the manuscript, only eribulin + OT-I treatment leads to significant survival advantage when compared to all other treatment groups.

8) No rationale provided for the various models used. Pancreatic neuroendocrine tumors, ovalbumin expressing TNBC and melanoma models, and human samples from HER2 negative cancers and rationale for dosing differences between drugs and between experiments not provided.

We consider it a strength of this manuscript that drug-induced pericyte phenotype switching was studied in 3 different cancer models. Pericyte phenotype switching and subsequent effects on tumor perfusion and/or hypoxia are consistent between models even though the microenvironment is different, thus demonstrating reproducibility and broad applicability of our approach.

Each of the models was chosen as outlined below and in the manuscript:

The RIP-Tag mouse model of pancreatic neuroendocrine tumors (PNET) is one of the best characterized models for tumor angiogenesis and has been widely studied from its inception in 1985 (Hanahan, 1985). It is highly predictive for the efficacy of anti-angiogenic drugs such as anti-VEGF/R treatments (Nowak-Sliwinska et al, 2018).

This is stated in Results (page 6):

“To assess the potency of microtubule-binding drugs to change pericyte phenotype *in vivo*, a highly angiogenic mouse pancreatic neuroendocrine tumor model (PNET, RIP1-Tag5) was employed which is predictive for the efficacy of anti-angiogenic drugs (Ganss & Hanahan, 1998; Nowak-Sliwinska et al, 2018).”

Most important for this study is the slow tumor growth kinetic which enables long-term treatment studies beyond the scope of implantation models. By using this transgenic model, we demonstrate the difference between anti-VEGFR treatment and its well-documented pro-metastatic effects (Casanovas et al. 2005) and eribulin/combretastatin treatments. It is a novel and crucial conclusion of our study that eribulin/combretastatin maintain vessel integrity over a prolonged period of time (8 weeks, Figure 4), which is not the case with anti-VEGF treatment. This is stated in Results (page 8):

“Treatments in the transgenic PNET model, due to its well characterized and slow tumor progression, can be monitored over a sustained period of time to address long-term drug effects beyond the limitations of implantation models (Casanovas et al, 2005).”

We do recognize, however, that eribulin is clinically approved for the treatment of advanced TNBC. We therefore included a mouse TNBC model. We selected this syngeneic C57BL/6 – CD45.2 model because it expresses a model tumor antigen (ovalbumin, OVA) which allows in depth immunological studies, and tracking of immune cells together with the congenic anti-OVA T cell receptor (TCR – CD45.1) transgenic mouse line.

This is stated in Results (page 9):

“To assess the stromal effects of these drugs on breast cancer, a C57BL/6 triple negative breast cancer (TNBC) model was employed which expresses the tumour antigen ovalbumin (AT3-OVA) for advanced immunological analyses in conjunction with congenic anti-OVA T cells (Dushyanthen et al, 2017).”

For instance, we were able to demonstrate that drug-induced vessel remodeling not only changes the tumor microenvironment but increases anti-tumor effector cell access. The Ova-TCR model is highly relevant for CAR T cell technology which is currently most successful in hematological malignancies, and would benefit from pre-treatment regimens as shown here in solid cancers. This is a major translational aspect of our study.

We also wanted to demonstrate that vessel normalization strategies which increase tumor perfusion and reduce hypoxia have different immune effects, depending on the pre-existing immune landscape. Therefore, we directly compared OVA-expressing breast cancer and melanoma models.

This is stated in Results (page 9):

“Although clinical applications of eribulin are thus far limited to advanced breast cancer and liposarcoma, it has shown therapeutic efficacy in human melanoma xenograft models (Asano

et al, 2018; Ito et al, 2017). We therefore also employed a B16-OVA (Falo et al, 1995) syngeneic mouse model to investigate pericyte phenotype switching and anti-tumor immunity in a different microenvironment.”

In breast cancer, T cells are major effectors. In contrast, in melanoma macrophage repolarization was a major feature associated with eribulin treatment.

Eribulin is clinically approved for advanced breast cancer. Therefore, eribulin-treated patients are normally pretreated with other drugs. Thus, pretreated clinical samples cannot be analyzed for eribulin-specific effects. Our patient cohorts of HER2-negative (TNBC and HR+) breast cancers represent a clinical trial of neoadjuvant eribulin (and no other treatments) where archival material is available for pre-, early and late eribulin treatments. This is a unique and valuable resource which allowed us to test longitudinal eribulin effects on two different breast cancer subtypes without interference of pre- or co-treatments.

To make it clearer that this patient cohort was not pre-treated, we added the following sentence to Results, page 11:

“This patient cohort enabled analysis of eribulin without other pre-treatments.”

That eribulin effects on the vasculature are comparable in both breast cancer subtypes is an interesting finding *per se*, in particular in light of the different responses to checkpoint inhibitor in TNBC and HR+ breast cancer patients, and our proposed scheduling for combination therapies.

The drug dosing for treatment is very consistent between mouse models:

RIP1-Tag5 (PNET): 1.5 mg/kg CA-4, 0.25 mg/kg eribulin, paclitaxel or vinorelbine

AT3-OVA (breast cancer): 1.5 mg/kg CA-4, 0.25 mg/kg eribulin, paclitaxel or vinorelbine

MO4-OVA (melanoma): 2.5 mg/kg CA-4, 0.25 mg/kg eribulin

9) *Abbreviation of combretastatin A4 is not consistent with the literature. Should be CA-4, not C-A4*

We have changed C-A4 to CA-4 to conform with the majority of publications as suggested.

10) *No data presented to support the overall conclusions that the pharmacological effects observed are mediated through ROCK signaling as all data are correlative. In the absence of*

this direct data, authors are advised to include references in the introduction describing the well-known effects of microtubule destabilizers on this pathway to provide additional rationale for the approach and conclusions.

We agree with Reviewer 1 that the data as presented are correlative.

To provide stronger evidence for causality in eribulin-mediated vessel normalization we have now included additional data in **revised Figure 2I** to demonstrate that suppression of Rho kinase *in vivo* (with fasudil) abolishes vascular effects of eribulin treatment, thus providing strong evidence for a causal role of Rho kinase in this process.

As stated above, effects of microtubule destabilizers on the Rho kinase pathway in pericytes (or even vSMC) are not “well known”, and our manuscript adds a new body of findings to the literature.

11) The authors need to more rigorously address the vascular stabilizing effects of CA-4 in their models/dosing regimens as compared to the vascular disrupting effects ascribed to this compound in the broader literature

We agree that the vascular stabilizing effects of low dose combretastatin reported here are interesting and novel. These effects are in stark contrast to the clinical development of high dose combretastatin as vascular disrupting agent. Our findings are particularly important since vascular disrupting compounds as single agents have been disappointing with neurotoxicity, cardiovascular and thromboembolic complications (Bates and Eastman, 2016), as now stated in the Introduction (page 4/5).

A substantial number of publications document endothelial cell death *in vitro* and tumour necrosis *in vivo* caused high dose combretastatin treatments.

Our manuscript makes significant and novel contributions by describing for the first time the vessel stabilizing phenomenon of low dose combretastatin treatment, and also delineates mechanistically the opposing roles of Rho kinase and ERK signaling in this process. As stated in the discussion, a more detailed analysis is indeed warranted and will be the subject of future publications (Discussion, page 13).

Referee #2 (Comments on Novelty/Model System for Author):

The role of pericyte maturation in anti-tumor immune response has been extensively studied. In the absence of novel mechanistic insights into the process of pericyte maturation the study appears to be confirmatory.

The novelty of our manuscript is the discovery that eribulin and combretastatin induce pericyte phenotype switching in the angiogenic vasculature. The ability to regulate pericyte contractile phenotype by re-dosing/re-scheduling microtubule-binding therapeutics is new and of high translational potential.

Reviewer 2 may refer to tumor blood vessel normalization/maturation (defined as an increase in pericyte numbers around vessels) which enhances immunotherapy, and is indeed a well-known phenomenon, so far mainly supported by genetic studies. There are currently no drugs which induce long-lasting vessel normalization and therefore clinical translation is lacking. Low dose anti-VEGF treatment normalizes vessels temporarily before chronic VEGF depletion leads to vessel death. The concept of pericyte phenotype switching in cancer has only recently been published by us, but not in the context of microtubule-binding drugs (Li et al, 2024). Therefore, this current manuscript provides the first mechanistic differentiation between vascular effects of low dose anti-VEGF and eribulin/combretastatin treatments.

That we improve immunotherapy following drug treatment is an important functional readout but does not constitute the novelty of the paper which is pericyte phenotype switching induced by combretastatin and eribulin.

Referee #2 (Remarks for Author):

The study by He et al. reports that the tubulin-binding drugs combretastatin A4 (C-A4) and eribulin induce tumor-associated pericyte maturation, enhance tumor perfusion, and improve responses to adoptive transfer or immune checkpoint blockade (ICB) therapy in animal models. The authors propose that C-A4 and eribulin achieve these effects by activating RhoA signaling and suppressing the ERK1/2 and AKT pathways, promoting a switch from the synthetic to the contractile phenotype of pericytes. They further demonstrate that contractile pericytes are enriched in human tumors following eribulin treatment, with a contractile signature associated with better treatment outcomes.

Overall, this is a technically solid and well-written paper that effectively delivers its message. While the concept of vascular, and specifically pericyte, maturation and its synergy with immunotherapy is not entirely novel, the novelty here lies in the specific effects of two tubulin-binding drugs on pericyte phenotypes. The demonstration of similar effects in human patient samples enhances the translational relevance of these findings. I have the following specific comments:

1. Do C-A4 and eribulin induce the same effects-such as pMLCK activation and pERK1/2 and mTOR inhibition-in angiogenic endothelial cells both in vitro and in vivo, and to what extent does this contribute to the tumor phenotypes observed in vivo?

New Figure EV1B demonstrates that pMLC is not upregulated in mouse endothelial cells (SVEC, SV40 transformed) following combretastatin, eribulin, paclitaxel or vinorelbine treatments at doses which induce pericyte phenotype switching *in vitro*.

Revised Figure 3 shows *in vivo* that endothelial-specific pERK or pS6R signaling is not suppressed in the same tumors which show drug-induced suppression in pericytes.

We therefore conclude that the observed low dose drug effects described in this manuscript directly change pericytes but not endothelial cells.

This is supported by Agoulnik et al, 2014, who found that eribulin but not paclitaxel specifically affected the gene expression profile of pericytes in an *in vitro* co-culture model with endothelial cells (see Discussion, page 13).

However, *in vivo*, it is likely that changes to one vascular population, pericytes or endothelial cells, will influence the whole vascular bed because of the intimate physical and chemical connections between the two cell types (Li et al, 2024).

Indeed, we demonstrated that VE-Cadherin expression is better aligned/stronger in eribulin/combretastatin-treated tumors (Figure 2F) which indicates improved stabilization of endothelial cells, most likely as an indirect effect of pericyte phenotype switching.

Since low dose eribulin/combretastatin treatments affected the entire vascular bed, including endothelial cells (Figure 2F), it was necessary for us to demonstrate in long-term treatments (Figure 4) that low dose drug applications did not induce endothelial cell death. This is an important finding which differentiates vessel normalization via pericytes from other treatment modalities which directly target endothelial cell, such as anti-VEGF therapy, and creates novel and more durable therapeutic opportunities.

2. *What is the proposed mechanism by which C-A4 and eribulin inhibit the synthetic phenotype? Specifically, how do these drugs activate RhoA, suppress pERK1/2 and the mTOR pathway, and induce the expression of contractile genes? Also, what is the impact of drugs on pericyte and endothelial cell proliferation?*

Proposed mechanism:

A prominent finding of our study is that the balance of two pathways, namely MEK/ERK and Rho kinase signaling, determines pericyte phenotype and vascular stability.

Signaling of microtubule-binding drugs in pericytes is an under-explored area.

From other cell types such as cancer cells, it is understood that microtubule-destabilizing drugs can upregulate Rho kinase via RhoGEFs (e.g, GEF-H1/ARHGEF2) which is directly associated with microtubules. GEF-H1 regulates Rho activity in response to microtubule depolymerization and links cellular actin polymerization and contractility to changes in microtubule dynamics (Krendel et al, 2002). How MEK/ERK/AKT signaling intersects with this pathway is not well understood and findings are controversial. There are several possibilities, which include downstream deactivation of ERK via GEF-H1 activation of JNK (Stone & Chambers, 2000). It is also possible that microtubule-destabilizing drugs directly regulated ERK and/or AKT signaling bypassing GEF-H1 (Jo et al, 2014; Nazmy et al, 2023).

In vascular smooth muscle cells (vSMC, close relatives to pericytes), seminal work by Owens and colleagues has shown that key regulatory kinases such as Rho kinase, MEK/ERK, and PI3K/AKT regulate the shift between contractile versus synthetic phenotypes directly via the actin network and at transcriptional level. This involves integration of receptor signaling with serum responsive factor (SRF) and CARrG box sequence in vSMC-specific genes (Owens, 2004; Tang et al, 2022; Frismantiene et al. 2018).

It is important to note that these signaling processes are well studied for vSMC for instance during wound healing; pericyte phenotype switching during tumor angiogenesis, however, is a new biological phenomenon which was only recently described by us (Li et al, 2024).

Detailed mechanistic insights into these complex signaling pathways are beyond the scope of this manuscript. However, we have included the following in the Discussion (page 13):

“How microtubule-targeting drugs induce pericyte phenotype switching involving Rho kinase and ERK and/or AKT signaling remains largely unknown. As shown in cancer cells, microtubule destabilizers such as CA-4 and eribulin can activate Rho kinase via the

microtubule-associated RhoGEF factor GEF-H1 (Krendel et al, 2002). This in turn may lead to ERK inactivation (Stone and Chambers, 2000). Alternatively, microtubule targeting may directly downregulated ERK and/or AKT signaling leading to Rho kinase activation (Jo et al, 2014; Nazmy et al, 2023; Wolfrum et al, 2004). In vascular smooth muscle cells, key regulatory kinases such as Rho kinase, MEK/ERK, and PI3K/AKT modulate phenotype directly via the actin cytoskeleton and at the transcriptional level (Lacolley et al, 2012).”

Pericyte and endothelial cell proliferation:

A considerable body of literature shows anti-proliferative properties of microtubule-binding drugs (including paclitaxel, combretastatin, eribulin) on endothelial cells *in vitro* as part of their anti-angiogenic activities in a highly dose-dependent manner (e.g. summarized in Schwartz et al, 2010).

We have previously shown *in vitro* that induction of pericyte contractile markers in 10T1/2 cells correlates with reduced cell cycle progression and proliferation (Li et al, 2024).

Assessment of pericyte or endothelial cell proliferation *in vivo* is challenging. Proliferation rates of vascular cells are exceedingly low under steady-state conditions; even under angiogenic stimulation *in vivo*, proliferation is much lower for vascular cells compared to cancer cells (Nowak-Liwinska et al, 2018). Therefore, commonly used immunohistochemistry markers such as Ki67 are too sparsely expressed in tumor vascular cells for quantitative analysis.

Overall vessel frequency is an approximation for vessel growth. Therefore, with each vascular marker assessment we also provided quantifications of CD31 (endothelial cell) and NG2 (pericyte) numbers (see Figures 2-5). These data show that endothelial/pericyte numbers remained constant between experimental groups demonstrating stable vessel numbers (indicating absence of proliferation or vessel loss).

In this context it is important to note that the contractile markers we identified for angiogenic pericytes are useful new markers for vessel normalization/quiescence which so far is difficult to assess *in vivo*.

3. The statement "eribulin induced contractile pericyte properties" lacks a functional readout to support this claim.

This is correct. We have changed this statement in Results (page 6) to “expression of contractile markers”, and all other references to contractility as indicated in the text.

4. The statement "*Mechanistically, C-A4 and eribulin both suppressed pERK signaling specifically in tumor pericytes (Fig. 3A)*" lacks sufficient specificity due to the absence of endothelial cell analysis.

Revised Figure 3 now includes endothelial cell analyses, and shows that eribulin/combretastatin effects are indeed specific for pericytes (PC, NG2⁺) but not endothelial cells (EC, CD31⁺).

5. To substantiate the statement, "*we demonstrated in mouse tumor models the selective capacity of low-dose C-A4 or eribulin to induce pericyte phenotype switching from a highly proliferative to a contractile state by activating RhoA kinase,*" the authors would need to either activate RhoA kinase in pericytes or show that inhibiting RhoA in pericytes prevents maturation in response to C-A4 and eribulin.

We agree. To demonstrate the key role of Rho kinase in modulating angiogenic pericytes/vessel status, we opted to inhibit Rho kinase with fasudil concomitant with eribulin treatment in PNET bearing mice (RIP1-Tag5 model).

Revised Figure 2I demonstrates that inhibition of Rho kinase during eribulin treatment abolishes the vessel improvement observed with eribulin (as measured by % increase in perfusion). This confirms a causal role for Rho kinase in mediating eribulin-induced vascular improvements.

Technical comments:

1. Figure 1 shows increases in *Cnn1*, *pMLCK*, and *ACTG2* in vitro upon treatment with C-A4 and eribulin, but not with other drugs. While the western blots are convincing, quantification is missing, and it is unclear how many times the experiment was repeated.

Revised Figure 1 now includes quantification of 3 independent experiments.

2. *pS6R* is a downstream target of the *mTOR* pathway. Although the *AKT* and *mTOR* pathways are mechanistically linked, they are not equivalent; using *pAKT* staining would provide a more accurate readout if the authors wish to emphasize *AKT* involvement.

We addressed Reviewer 2's concerns regarding potential integration of PI3K/AKT/mTor signaling as follows:

pAKT: we analyzed pericytes in untreated and eribulin treated tumors for both pS6R (Figure 2) and pAKT activity (**new Figure EV3A**) with the same outcome, namely suppression of pAKT signaling with eribulin treatment, specifically confirming involvement of AKT signaling.

New Figures EV3B and C also demonstrate that Wortmannin (PI3K inhibitor which downregulates pAKT, see Western blot control in B) upregulates contractile pericyte markers *in vitro* and improves vascular function *in vivo*, thus demonstrating involvement of PI3K/AKT in pericyte phenotype switching.

Minor comment: In "vascular pericytes," remove "vascular."

This has been removed on page 6 of the manuscript.

Referee #3 (Comments on Novelty/Model System for Author):

This paper reports an interesting finding that has a potential for a high translational impact. The study is technically sound, I see no obvious issues.

Referee #3 (Remarks for Author):

The manuscript presents a finding that low concentrations of microtubule-binding drugs CA4 and eribulin induce pericyte differentiation by inducing RhoA activation, leading to vasculature normalization. In turn, vasculature normalization reduced hypoxia, linked with more aggressive phenotypes of tumor cells, and enhanced the accessibility and effects of immune therapies.

The data linking tubulin-binding drugs with improved vascularization and enhanced therapeutic outcomes is compelling. The manuscript is well-written, and the data is logically and clearly presented. The reported findings have a high translational potential.

My only non-critical concern (as it does not really impact the translational potential of the reported findings) is about the inferences of the mechanisms, which is postulated to involve RhoA activation caused by simultaneous suppression of the MEK/ERK and AKT/PI3K. While plausible, the manuscript misses an obvious opportunity to validate their mechanistic inferences using RhoA inhibitor to suppress the effect of epirubicin and CA4, or a combination of MEK/AKT inhibitors to phenocopy the effect. Achieving the expected effect would strongly strengthen the mechanistic inferences, while the inability to achieve the effects would indicate the involvement of other or additional mechanisms, which would not compromise the impact of the study.

As suggested by Reviewer 3, we have further investigated the ability of various pathway inhibitors to interfere with the vascular stabilizing effects of eribulin *in vitro* and *in vivo*.

Revised Figure 2I demonstrates that Rho kinase upregulation is crucial for vascular remodeling since specific Rho kinase inhibition by fasudil (small molecule inhibitor of Rho kinase) abolishes eribulin effects on tumor perfusion (please also see also response to Reviewer 1 and 2).

New Figure EV3B demonstrates *in vitro* that inhibition of ERK and/or PI3K pathways using the inhibitors PD98059 or Wortmannin, respectively, increases contractile markers (ACTG2

and pMLC) in angiogenic pericytes, similar to treatment with eribulin or combretastatin (Figure 1). Simultaneous treatment with both inhibitors increases the effects, demonstrating synergy in these pathways.

In addition, **new Figure EV3C** demonstrates in PNET-bearing mice *in vivo* that blocking of ERK (PD98059) or IP3K (Wortmannin) signaling improves vascular function as measured by increased tumor perfusion.

Please note that we also tested the efficacy of dual treatment (PD98059 and Wortmannin) in PNET, however, double treatment reduces vessel viability *in vivo* (data not shown).

Overall, these data strongly support our conclusion that pericyte maturity/vessel stabilization are mediated via Rho kinase upregulation and suppression of ERK/IP3K pathways.

• Another obvious question is how microtubule binding drugs impact the observed signaling changes, and what could be the mechanisms responsible for the pericyte specificity. While addressing these questions might be outside of the scope of the paper, I suggest at least speculating on it in the discussion or explicitly mentioning the lack of knowledge about the mechanisms underlying the connection.

Please also see our response to Reviewer 2/Point 2.

We have now included a discussion on the potential mechanism in our revised manuscript (page 13).

“How microtubule-targeting drugs induce pericyte phenotype switching involving Rho kinase and ERK and/or AKT signaling remains largely unknown. As shown in cancer cells, microtubule destabilizers such as CA-4 and eribulin can activate Rho kinase via the microtubule-associated RhoGEF factor GEF-H1 (Krendel et al, 2002). This in turn may lead to ERK inactivation (Stone and Chambers, 2000). Alternatively, microtubule targeting may directly downregulated ERK and/or AKT signaling leading to Rho kinase activation (Jo et al, 2014; Nazmy et al, 2023; Wolfrum et al, 2004; Mavria et al, 2006). In vascular smooth muscle cells, key regulatory kinases such as Rho kinase, MEK/ERK, and PI3K/AKT modulate phenotype directly via the actin cytoskeleton and at the transcriptional level (Lacolley et al, 2012).”

• While not essential, it would be highly desirable to validate vasculature normalization with an alternative vasculature-specific stain such as CD34 or MECA32, especially for the clinical samples where experimental approaches to characterize vasculature integrity and

hypoxia are not accessible. That is if the authors have access to additional tissue sections from the trial.

Vascular markers such as CD31 (shown in Figure 8) or indeed CD105 or CD34 (MECA32 does not cross-react with human blood vessels) do not *per se* show vessel normalization effects.

To address Reviewer 3's concerns regarding an additional marker for vessel integrity, we have stained patient breast cancer biopsies with VE-Cadherin. VE-Cadherin is an endothelial-cell specific adhesion molecule. Its expression pattern along the vascular bed is a marker for the integrity of the endothelial cell barrier, and indeed the vascular bed.

Revised Figure 8 shows that VE-Cadherin signal continuity along CD31⁺ endothelial cells is improved with eribulin treatment in human breast cancer demonstrating improved endothelial cell/vascular integrity. This is consistent with our mouse tumor data shown in Figure 2.

• Finally, it would be desirable to provide evidence of RhoA activation and vasculature normalization for the B16-OVA model used in the adoptive T cell transfer study reported in Figure 7.

Figure EV4 shows pericyte-specific pMLC and ACTG2 expression in B16-OVA tumors treated with eribulin/combretastain, as well as tumor perfusion and hypoxia data. These vessel normalization data complement immune data shown in Figure 7.

Minor points:

• A scale bar needs to be provided for the Figure 8A data.

We have now inserted a scale bar in each row of Figure 8A.

• Page 7 "inducing higher and more coherent expression of ...VE-cadherin" - what does "more coherent expression" mean? Consider explaining or rephrasing.

In the context of VE-Cadherin expression on endothelial cells "more coherent" means a less disrupted, more continuous expression pattern which correlates with higher vessel integrity/less permeability and improved barrier function.

To make this clearer to the reader, we have amended the text in Results (page 7) as follows:

“Moreover, CA-4 or eribulin treatment had significant effects on the entire vascular bed by inducing higher and more **continuous** expression of the vascular adhesion marker VE-cadherin **implying improved barrier function.**”

- *Consider providing volumetric traces for individual tumors in the supplementary information.*

Appendix Figure S1 now shows individual tumor size traces for survival studies on AT3-OVA breast cancer (related to Figure 6) and B16-OVA melanoma (related Figure 7).

- *In the Results section, the authors refer to a non-existing panel 6H instead of the 6G, this needs to be fixed.*

This has been amended.

20th Feb 2025

Dear Prof. Ganss,

Thank you for submitting your revised study. We have now received the reports from the three referees who evaluated your revised manuscript. As you will see from the reports below, they are satisfied with the revisions, and I will therefore be able to accept your manuscript once the following editorial issues are addressed:

1/ Please address the remaining minor comments from the referees.

2/ Manuscript text:

- Remove the blue font and only keep in track changes mode any new modification.
- Please provide up to 5 keywords.
- Methods:
 - o Patients and human tissue: Include a statement confirming that informed consent was obtained from all subjects and that the experiments conformed to the principles set out in the WMA Declaration of Helsinki and the Department of Health and Human Services Belmont Report.
 - o Cells: please indicate whether the cells were authenticated and tested for mycoplasma contamination.
 - o Antibodies: please provide dilutions/concentrations.
 - o Statistical analysis: please provide a statement on sample size and inclusion/exclusion criteria.
- Data availability: Please remove the sentence "The data generated in this study are included in source data or available upon request from the corresponding author."
- The disclosure and competing interests statements should be placed after the Acknowledgements. Please add the following sentence: "RG is an editorial advisory board member."
- Acknowledgements: The information provided in the manuscript should match the information provided in the submission system (currently, Cancer Australia is P0411 in the ms vs. PO411 in the system).

3/ Figures and Appendix:

- Please note that we can now accommodate more than 5 EV figures, in case you would like to make some of your Appendix figures more accessible to the readers.
- Please make sure that all figures/figure panels are referenced in the manuscript text (currently, a callout is missing for Fig. 1E). The following callouts need to be corrected: Supplementary Table S1-S3.
- During our standard figure check, we noted that Figure 1C appears to have a splice site in the figure AND in the source data. Please check, clarify and correct if needed.
- Please address the queries from our copy editors:
 1. Please note that the exact p values are not provided in the legends of figures 2G, 3B, 4D, E; 5F, 6B, F; 7C, 8B, C.
 2. Please note that the error bars are not defined in the legends of figures 8B, C.
 3. Please note that the white arrows are not defined in the legend of figure 3B, 4B, 8A, EV3 A. This needs to be rectified.
 4. Please note that the dotted borders are not defined in the legend of figure 4E. This needs to be rectified.

4/ Checklist:

- Please fill in the entire "Experimental study design and statistics" section.
- Please fill in the subsection on 'informed consent' & 'Declaration of Helsinki' in the "Ethics" section.

5/ Thank you for providing The paper explained. Please include it in the manuscript file.

6/ Synopsis:

I have made minor edits in your synopsis, please let me know if you agree with the following or amend as you see fit:

"Selective microtubule-binding drugs (i.e. eribulin) induce phenotype switching of angiogenic pericytes, which profoundly changes the tumor microenvironment. Re-dosing/re-scheduling of established drugs creates new therapeutic opportunities for cancers intrinsically resistant to immunotherapy.

- Drug-induced pericyte phenotype switching is regulated by signaling pathways converging on Rho kinase activity.
- Drug-induced pericyte maturation normalizes tumor blood vessels, improves tumor oxygenation and creates an immune-supportive environment.
- Vessel normalization via pericyte maturation is highly durable and - in contrast to anti-angiogenic VEGF targeting - does not induce tumor metastasis.
- Pericyte phenotype switching is a direct result of eribulin neoadjuvant treatment in breast cancer patients.
- Pericyte "maturity" signatures correlate with improved survival in breast cancer patients."

Thank you for providing a nice synopsis image. I have cropped a small portion (115x70 pixels) to serve as a thumbnail on our website (attached). Please let me know if you agree or would rather provide an alternative image.

7/ As part of the EMBO Publications transparent editorial process initiative (see our Editorial at <http://embomolmed.embopress.org/content/2/9/329>), EMBO Molecular Medicine will publish online a Review Process File (RPF) to accompany accepted manuscripts.

This file will be published in conjunction with your paper and will include the anonymous referee reports, your point-by-point response and all pertinent correspondence relating to the manuscript. Let us know whether you agree with the publication of the RPF.

I look forward to receiving your revised manuscript.

With kind regards,

Lise

**** Reviewer's comments ****

Referee #1 (Remarks for Author):

Significant improvements have been made to the presentation of data and incorporation of findings into the broader literature. This is a robust study demonstrating the ability of some, but not all MTAs, to promote vascular remodeling to improve immune infiltration through RhoA activation in pericytes with in vitro, in vivo (multiple models), and clinical data.

Two minor comments on wording included below:

- New language added to the manuscript includes the full name 'combretastatin' as opposed to the CA-4 abbreviation used in the rest of the manuscript
- The interaction of ERK and PI3K inhibition appears to be an additive effect at the single concentration used as opposed to a synergistic effect as described (this does not diminish significance of the finding but is just a more accurate description of the finding)

Referee #2 (Remarks for Author):

The authors answered most of the critiques constructively. I am a bit surprised by staining pattern for pAKT which could be interpreted as pAKT + immune cells associated with pericytes, difficult to say at this magnification.

Referee #3 (Remarks for Author):

The authors have adequately addressed the comments/suggestions. I have no additional suggestions/requests.

HARRY PERKINS INSTITUTE
OF MEDICAL RESEARCH

Dr Lise Roth
Senior Editor
EMBO Molecular Medicine

February 24, 2025

Dear Lise,

RE: Manuscript EMM-2024-20709v2

Thank you for the opportunity to further revise our manuscript entitled “**Selective tubulin-binding drugs induce pericyte phenotype switching and anti-cancer immunity**” by Bo He et al.

In response to your recent communication from February 20 the following was actioned:

1. We have answered the reviewers' remaining concerns in the enclosed point-by-point response, modified the text and provide a **revised Figure EV3A**.
2. Manuscript text:
We have made changes in the text as requested, all highlighted in green “track-changes”. Specifically, antibody dilutions are now shown in Methods (text) and in **revised Appendix Table S2**.
3. Figures/Appendix:
To expand our EV Figures, we have added a **new Figure EV6** (formerly Appendix Figure S2) to be shown in conjunction with Figure 8. The Appendix was revised accordingly.
Figure 1 C: we acknowledge your integrity screening results, but can confirm that the images were generated from one gel, not spliced together. Nevertheless, we have now provided in **revised Figure 1C** the Western blot images from our repeat experiment 3 (previous Figure 1C was experiment 1). The corresponding source file data contain all three of our quantified experiments. We hope that this new image 1C will pass your screening test, but all Western blots shown in Figure 1 B-E were generated in the same way, with dose escalation data of one drug run on the same gel.
Please note: All p values displayed as **** $p < 0.0001$ in figure legends 2G; 3B; 4D, E; 5F; 6B, F; 7C; 8B,C are the smallest values our statistics program determines. We can't provide smaller p values.
We have now added to Fig. 8B Figure legend that “Individual patient data are shown.” This is why there are no error bars.

4. Checklist:
A revised checklist was uploaded.
5. "Paper explained" is now included in the main manuscript.
6. Synopsis edits and thumbnail are fine.
7. We agree with publication of the RPF.

Thank you again for considering our revised manuscript.

Yours sincerely,

Prof. Ruth Ganss
Harry Perkins Institute of Medical Research,
Perth, Western Australia, Australia
Phone: ++ 61 8 6151 0733
E-MAIL: ganss@perkins.org.au

Point-by-point rebuttal, revision 2

Please find responses to Reviewer comments below.

Referee #1 (Remarks for Author):

Significant improvements have been made to the presentation of data and incorporation of findings into the broader literature. This is a robust study demonstrating the ability of some, but not all MTAs, to promote vascular remodeling to improve immune infiltration through RhoA activation in pericytes with in vitro, in vivo (multiple models), and clinical data.

Two minor comments on wording included below:

- New language added to the manuscript includes the full name 'combretastatin' as opposed to the CA-4 abbreviation used in the rest of the manuscript

We have corrected this and exchanged combretastatin for CA-4 on pages 5, 6 and 8.

- The interaction of ERK and PI3K inhibition appears to be an additive effect at the single concentration used as opposed to a synergistic effect as described (this does not diminish significance of the finding but is just a more accurate description of the finding)

We agree with this comment based on the additive effects of ERK and PI3K pathway inhibition *in vitro* as shown in Figure EV3. We have changed the text as follows in Results, page 6:

Simultaneous treatment with both inhibitors increases expression, implying additive effects of these pathways (Fig. EV3B).

Referee #2 (Remarks for Author):

The authors answered most of the critiques constructively. I am a bit surprised by staining pattern for pAKT which could be interpreted as pAKT + immune cells associated with pericytes, difficult to say at this magnification.

pAKT as shown in Fig. EV3A (and also pS6R in Fig. 3) stains mainly proliferating cancer cells in tumors which are stained in red in the shown microscopic images. What was quantified -and is the focus of our study- is the vascular expression of pAKT; these signals appear in yellow when pAKT (red) is overlaid with a pericyte marker (NG2, green), and are highlighted by arrows. Because of the density of “red” cancer signals in the image in current Fig. EV3A (“untreated”), it may have appeared that cells close to vessels are pAKT positive. We have now replaced the “untreated” image (left) in **revised Fig. EV3A** with a representative image displaying less pAKT⁺ cancer cells to show more prominently vessel-intrinsic pAKT signals.

Referee #3 (Remarks for Author):

The authors have adequately addressed the comments/suggestions. I have no additional suggestions/requests.

No further modifications required.

28th Feb 2025

Dear Prof. Ganss,

Thank you for submitting your revised files. I am pleased to inform you that your manuscript is accepted for publication and is now being sent to our publisher to be included in the next available issue of EMBO Molecular Medicine!

With kind regards,

Lise
